# A Conditional Point Diffusion-Refinement Paradigm for 3D Point Cloud Completion

**Zhaoyang Lyu**[1,2*] **Zhifeng Kong**[3*] **Xudong Xu**[1] **Liang Pan**[4] **Dahua Lin**[1,2]

[1]CUHK-SenseTime Joint Lab, The Chinese University of Hong Kong
[2]Shanghai AI Laboratory    [3]University of California, San Diego
[4]S-Lab, Nanyang Technological University
lyuzhaoyang@link.cuhk.edu.hk, z4kong@eng.ucsd.edu
xx018@ie.cuhk.edu.hk, liang.pan@ntu.edu.sg, dhlin@ie.cuhk.edu.hk

## Abstract

3D point cloud is an important 3D representation for capturing real world 3D objects. However, real-scanned 3D point clouds are often incomplete, and it is important to recover complete point clouds for downstream applications. Most existing point cloud completion methods use Chamfer Distance (CD) loss for training. The CD loss estimates correspondences between two point clouds by searching nearest neighbors, which does not capture the overall point density distribution on the generated shape, and therefore likely leads to non-uniform point cloud generation. To tackle this problem, we propose a novel **Point Diffusion-Refinement (PDR) paradigm** for point cloud completion. PDR consists of a Conditional Generation Network (CGNet) and a ReFinement Network (RFNet). The CGNet uses a conditional generative model called the denoising diffusion probabilistic model (DDPM) to generate a coarse completion conditioned on the partial observation. DDPM establishes a one-to-one pointwise mapping between the generated point cloud and the uniform ground truth, and then optimizes the mean squared error loss to realize uniform generation. The RFNet refines the coarse output of the CGNet and further improves quality of the completed point cloud. Furthermore, we develop a novel dual-path architecture for both networks. The architecture can (1) effectively and efficiently extract multi-level features from partially observed point clouds to guide completion, and (2) accurately manipulate spatial locations of 3D points to obtain smooth surfaces and sharp details. Extensive experimental results on various benchmark datasets show that our PDR paradigm outperforms previous state-of-the-art methods for point cloud completion. Remarkably, with the help of the RFNet, we can accelerate the iterative generation process of the DDPM by up to 50 times without much performance drop.

## 1 Introduction

With the rapid developments of 3D sensors, 3D point clouds are an important data format that captures 3D information owing to their ease of acquisition and efficiency in storage. Unfortunately, point clouds scanned in the real world are often incomplete due to partial observation and self occlusion. It is important to recover the complete shape by inferring the missing parts for many downstream tasks such as 3D reconstruction, augmented reality and scene understanding. To tackle this problem, many learning-based methods (Yuan et al., 2018; Yang et al., 2018; Tchapmi et al., 2019; Xie et al., 2020; Liu et al., 2020; Pan et al., 2021) are proposed, which are supervised by using either the Chamfer Distance (CD) or Earth Mover Distance (EMD) to penalize the discrepancies between the generated complete point cloud and the ground truth. However, CD loss is not sensitive to overall density distribution, and thus networks trained by CD loss could generate non-uniform point cloud completion results (See Figure 10 and 11 in Appendix). EMD is more distinctive to measure density distributions, but it is too expensive to compute in training. The absence of an effective and efficient training loss highly limits the capabilities of many existing point cloud completion networks.

---

[*]Equal Contribution.   Code is released at https://github.com/ZhaoyangLyu/Point_Diffusion_Refinement.

Figure 1: Our Conditional **P**oint **D**iffusion-**R**efinement (PDR) paradigm first moves a Gaussian noise step by step towards a coarse completion of the partial observation through a diffusion model (DDPM). Then it refines the coarse point cloud by one step to obtain a high quality point cloud.

We find that denoising diffusion probabilistic models (DDPM) (Sohl-Dickstein et al., 2015; Ho et al., 2020) can potentially generate uniform and high quality point clouds with an effective and efficient loss function. It can iteratively move a set of Gaussian noise towards a complete and clean point cloud. DDPM defines a one-to-one pointwise mapping between two consecutive point clouds in the diffusion process, which enables it to use a simple mean squared error loss function for training. This loss function is efficient to compute and explicitly requires the generated point cloud to be uniform, as a one-to-one point mapping is naturally established between the generated point cloud and the ground truth. Point cloud completion task can be treated as a conditional generation problem in the framework of DDPM (Zhou et al., 2021; Luo & Hu, 2021). Indeed, we find the complete point clouds generated by a conditional DDPM often have a good overall distribution that uniformly covers the shape of the object.

Nonetheless, due to the probabilistic nature of DDPM and the lack of a suitable network architecture to train the conditional DDPM for 3D point cloud completion in previous works, we find DDPM completed point clouds often lack smooth surfaces and sharp details (See Figure 1 and Appendix Figure 12), which is also reflected by their high CD loss compared with state-of-the-art point cloud completion methods in our experiments. Another problem with DDPM is its inefficiency in the inference phase. It usually takes several hundreds and even up to one thousand forward steps to generate a single point cloud. Several methods (Song et al., 2020; Nichol & Dhariwal, 2021; Kong & Ping, 2021) are proposed to accelerate DDPM using jumping steps without retraining networks, which however, leads to an obvious performance drop when using a small number of diffusion steps.

In this work, we propose the Conditional **P**oint **D**iffusion-**R**efinement (**PDR**) paradigm to generate both uniform and high quality complete point clouds. As shown in Figure 1, our PDR paradigm performs point cloud completion in a coarse-to-fine fashion. Firstly, we use the Conditional Generation Network (CGNet) to generate a coarse complete point cloud by the DDPM conditioned on the partial point cloud. It iteratively moves a set of Gaussian noise towards a complete point cloud. Following, the ReFinement Network (RFNet) further refines the coarse complete point cloud generated from the Conditional Generation Network with the help of partial point clouds. In addition, RFNet can be used to refine the low quality point clouds generated by an accelerated DDPM, so that we could enjoy an acceleration up to 50 times, while minimizing the performance drop. In this way, the completion results generated by our PDR paradigm demonstrate both good overall density distribution (*i.e.* uniform) and sharp local details.

Both CGNet and RFNet have a novel dual-path network architecture shown in Figure 2, which is composed of two parallel sub-networks, a Denoise subnet and a Condition Feature Extraction subnet for noisy point clouds and partial point clouds, respectively. Specifically, we propose Point Adaptive Deconvolution (PA-Deconv) operation for upsampling, which can effectively manipulate spatial locations of 3D points. Furthermore, we propose the Feature Transfer (FT) module to directly transmit encoded point features at different scales from the Condition Feature Extraction subnet to the corresponding hierarchy in the Denoise subnet. Extensive experimental results show that our PDR paradigm can provide new state-of-the-art performance for point cloud completion.

Our **Key** contributions can be summarized as: **1)** We identify conditional DDPM to be a good model with an effective and efficient loss function to generate uniform point clouds in point cloud completion task. **2)** By using RFNet to refine the coarse point clouds, our PDR paradigm can generate complete point cloud with both good overall density distribution (*i.e.* uniform) and sharp local details. **3)** We design novel point learning modules, including PA-Deconv and Feature Transfer modules, for constructing CGNet in DDPM and RFNet, which effectively and efficiently utilizes multi-level features extracted from incomplete point clouds for point cloud completion. **4)** With the help of our proposed RFNet, we can accelerate the generation process of DDPM up to 50 times without a significant drop in point cloud quality.

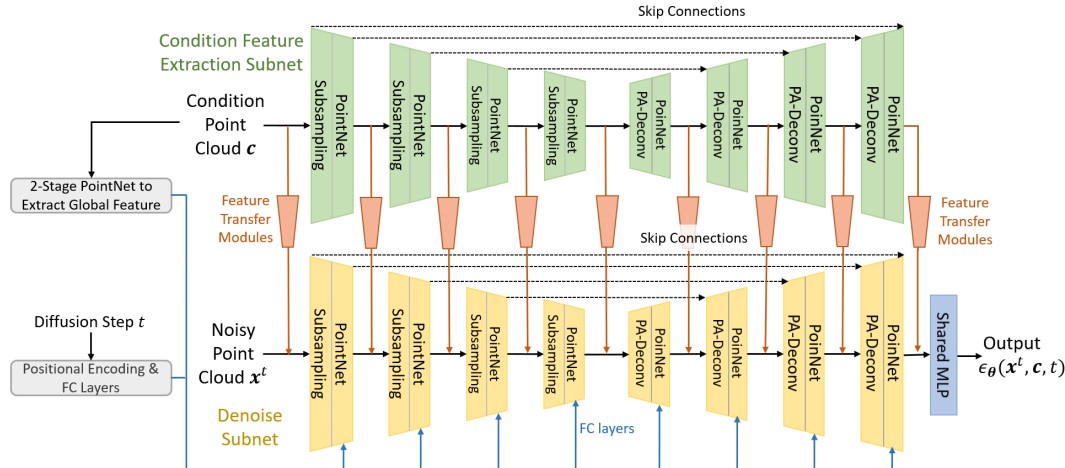

Figure 2: Network architecture of the Conditional Generation Network (CGNet) and ReFinement Network (RFNet). It consists of the Condition Feature Extraction subnet and the Denoise subnet.

## 2 PROBLEM STATEMENT

In this paper, we focus on the 3D point cloud completion task. A 3D point cloud is represented by $N$ points in the 3D space: $\boldsymbol{X} = \{x_j | 1 \leq j \leq N\}$, where each $x_j \in \mathbb{R}^3$ is the 3D coordinates of the $j$-th point. We assume the dataset is composed of $M$ data pairs $\{(\boldsymbol{X}_i, \boldsymbol{C}_i) | 1 \leq i \leq M\}$, where $\boldsymbol{X}_i$ is the $i$-th ground-truth point cloud, and $\boldsymbol{C}_i$ is the incomplete point cloud from a partial observation of $\boldsymbol{X}_i$. The goal is to develop a model that completes the partial observation $\boldsymbol{C}_i$ and outputs a point cloud as close to the ground truth $\boldsymbol{X}_i$ as possible. For algebraic convenience, we let $\boldsymbol{x} \in \mathbb{R}^{3N}$ be the vector form of a point cloud $\boldsymbol{X}$, and similarly $\boldsymbol{c}$ be the vector form of $\boldsymbol{C}$.

## 3 METHODOLOGY

We consider the point cloud completion task as a conditional generation problem, where the incomplete point cloud $\boldsymbol{C}$ serves as the conditioner. We use the powerful generative model called denoising diffusion probabilistic models (DDPM) (Sohl-Dickstein et al., 2015; Ho et al., 2020; Kong et al., 2020) to first generate a coarse completion of the partial observation. Then we use another network to refine the coarse point cloud to improve its visual quality. Our point cloud completion pipeline is shown in Figure 1. We first briefly introduce the theory of DDPM in Section 3.1, and then describe detailed architecture of the Conditional Generation Network (CGNet) and ReFinement Network (RFNet) in Section 3.2 and Section 3.3.

### 3.1 BACKGROUND ON CONDITIONAL DENOISING DIFFUSION PROBABILISTIC MODELS

We assume $p_{\text{data}}$ to be the distribution of the complete point cloud $\boldsymbol{x}_i$ in the dataset, and $p_{\text{latent}} = \mathcal{N}(\boldsymbol{0}_{3N}, \boldsymbol{I}_{3N \times 3N})$ to be the latent distribution, where $\mathcal{N}$ is the Gaussian distribution. Then, the conditional DDPM consists of two Markov chains called the diffusion process and the reverse process. Both processes have length equal to $T$. We set $T = 1000$ in this paper.

**The Diffusion Process.** The diffusion process is a Markov process that adds Gaussian noise into the clean data distribution $p_{\text{data}}$ until the output distribution is close to $p_{\text{latent}}$. The diffusion process is irrelevant of the conditioner, the incomplete point cloud $\boldsymbol{c}_i$. Formally, let $\boldsymbol{x}^0 \sim p_{\text{data}}$. We use the superscript to denote the diffusion step $t$. For conciseness, we omit the subscription $i$ in the following discussion. The diffusion process from clean data $\boldsymbol{x}^0$ to $\boldsymbol{x}^T$ is defined as

$$q(\boldsymbol{x}^1, \cdots, \boldsymbol{x}^T | \boldsymbol{x}^0) = \prod_{t=1}^{T} q(\boldsymbol{x}^t | \boldsymbol{x}^{t-1}), \text{ where } q(\boldsymbol{x}^t | \boldsymbol{x}^{t-1}) = \mathcal{N}(\boldsymbol{x}^t; \sqrt{1 - \beta_t} \boldsymbol{x}^{t-1}, \beta_t \boldsymbol{I}). \quad (1)$$

The hyperparameters $\beta_t$ are pre-defined, small positive constants (See details in Appendix Section A.1). According to Ho et al. (2020), there is a closed form expression for $q(\boldsymbol{x}^t | \boldsymbol{x}^0)$. We first define constants $\alpha_t = 1 - \beta_t$, $\bar{\alpha}_t = \prod_{i=1}^{t} \alpha_i$. Then, we have $q(\boldsymbol{x}^t | \boldsymbol{x}^0) = \mathcal{N}(\boldsymbol{x}^t; \sqrt{\bar{\alpha}_t} \boldsymbol{x}^0, (1 - \bar{\alpha}_t) \boldsymbol{I})$.

Therefore, when $T$ is large enough, $\bar{\alpha}_t$ goes to 0, and $q(\boldsymbol{x}^T|\boldsymbol{x}^0)$ becomes close to the latent distribution $p_{\text{latent}}(\boldsymbol{x}^T)$. Note that $\boldsymbol{x}^t$ can be directly sampled through the following equation:

$$\boldsymbol{x}^t = \sqrt{\bar{\alpha}_t}\boldsymbol{x}^0 + \sqrt{1 - \bar{\alpha}_t}\boldsymbol{\epsilon}, \text{ where } \boldsymbol{\epsilon} \text{ is a standard Gaussian noise.} \tag{2}$$

We emphasize that $q(\boldsymbol{x}^t|\boldsymbol{x}^{t-1})$ can be seen as a one-to-one pointwise mapping as $\boldsymbol{x}^t$ can be sampled through the equation $\boldsymbol{x}^t = \sqrt{1 - \beta_t}\boldsymbol{x}^{t-1} + \beta_t\boldsymbol{\epsilon}$. Therefore, the order of points in $\boldsymbol{x}^0$ is preserved in the diffusion process. However, it does not matter what kind of order we input the points in $\boldsymbol{x}^0$. That is because when $T$ is large enough, $\boldsymbol{x}^T$ will become a Gaussian distribution. Every point in a Gaussian distribution is equivalent and there is no way to distinguish one point from another.

**The Reverse Process.** The reverse process is a Markov process that predicts and eliminates the noise added in the diffusion process. The reverse process is conditioned on the conditioner, the incomplete point cloud $\boldsymbol{c}$. Let $\boldsymbol{x}^T \sim p_{\text{latent}}$ be a latent variable. The reverse process from latent $\boldsymbol{x}^T$ to clean data $\boldsymbol{x}^0$ is defined as

$$p_{\boldsymbol{\theta}}(\boldsymbol{x}^0, \cdots, \boldsymbol{x}^{T-1}|\boldsymbol{x}^T, \boldsymbol{c}) = \prod_{t=1}^{T} p_{\boldsymbol{\theta}}(\boldsymbol{x}^{t-1}|\boldsymbol{x}^t, \boldsymbol{c}), \text{ where } p_{\boldsymbol{\theta}}(\boldsymbol{x}^{t-1}|\boldsymbol{x}^t, \boldsymbol{c}) = \mathcal{N}(\boldsymbol{x}^{t-1}; \boldsymbol{\mu}_{\boldsymbol{\theta}}(\boldsymbol{x}^t, \boldsymbol{c}, t), \sigma_t^2\boldsymbol{I}).$$
$$\tag{3}$$

The mean $\boldsymbol{\mu}_{\boldsymbol{\theta}}(\boldsymbol{x}^t, \boldsymbol{c}, t)$ is a neural network parameterized by $\boldsymbol{\theta}$ and the variance $\sigma_t^2$ is a time-step dependent constant. To generate a sample conditioned on $\boldsymbol{c}$, we first sample $\boldsymbol{x}^T \sim \mathcal{N}(\boldsymbol{0}_{3N}, \boldsymbol{I}_{3N \times 3N})$, then draw $\boldsymbol{x}^{t-1} \sim p_{\boldsymbol{\theta}}(\boldsymbol{x}^{t-1}|\boldsymbol{x}^t, \boldsymbol{c})$ for $t = T, T-1, \cdots, 1$, and finally outputs $\boldsymbol{x}^0$.

**Training.** DDPM is trained via variational inference. Ho et al. (2020) introduced a certain parameterization for $\boldsymbol{\mu}_{\boldsymbol{\theta}}$ that can largely simplify the training objective. The parameterization is $\sigma_t^2 = \frac{1-\bar{\alpha}_{t-1}}{1-\bar{\alpha}_t}\beta_t$, and $\boldsymbol{\mu}_{\boldsymbol{\theta}}(\boldsymbol{x}^t, \boldsymbol{c}, t) = \frac{1}{\sqrt{\alpha_t}}\left(\boldsymbol{x}^t - \frac{\beta_t}{\sqrt{1-\bar{\alpha}_t}}\boldsymbol{\epsilon}_{\boldsymbol{\theta}}(\boldsymbol{x}^t, \boldsymbol{c}, t)\right)$, where $\boldsymbol{\epsilon}_{\boldsymbol{\theta}}$ is a neural network taking noisy point cloud $\boldsymbol{x}^t \sim q(\boldsymbol{x}^t|\boldsymbol{x}^0)$ in equation (2), diffusion step $t$, and conditioner $\boldsymbol{c}$ as inputs. Then, the simplified training objective becomes

$$L(\boldsymbol{\theta}) = \mathbb{E}_{i \sim \mathcal{U}([M]), t \sim \mathcal{U}([T]), \boldsymbol{\epsilon} \sim \mathcal{N}(0,I)} \|\boldsymbol{\epsilon} - \boldsymbol{\epsilon}_{\boldsymbol{\theta}}(\sqrt{\bar{\alpha}_t}\boldsymbol{x}_i^0 + \sqrt{1 - \bar{\alpha}_t}\boldsymbol{\epsilon}, \boldsymbol{c}_i, t)\|^2, \tag{4}$$

where $\mathcal{U}([M])$ is the uniform distribution over $\{1, 2, \cdots, M\}$. The neural network $\boldsymbol{\epsilon}_{\boldsymbol{\theta}}$ learns to predict the noise $\boldsymbol{\epsilon}$ added to the clean point cloud $\boldsymbol{x}^0$, which can be used to denoise the noisy point cloud $\boldsymbol{x}^t = \sqrt{\bar{\alpha}_t}\boldsymbol{x}^0 + \sqrt{1 - \bar{\alpha}_t}\boldsymbol{x}^0$. Note that traditional CD loss or EMD loss is NOT present in Equation 4. The reason that we are able to use the simple mean squared error is because DDPM naturally defines a one-to-one pointwise mapping between two consecutive point clouds in the diffusion process as shown in Equation 1. Note that at each training step, we not only need to sample a pair of point clouds $\boldsymbol{x}_i, \boldsymbol{c}_i$, but also a diffusion step $t$ and a Gaussian noise $\boldsymbol{\epsilon}$.

## 3.2 CONDITIONAL GENERATION NETWORK

In this section, we introduce the architecture of Conditional Generation Network (CGNet) $\boldsymbol{\epsilon}_{\boldsymbol{\theta}}$. The inputs of this network are the noisy point cloud $\boldsymbol{x}^t$, the incomplete point cloud $\boldsymbol{c}$, and the diffusion step $t$. We can intuitively interpret the output of $\boldsymbol{\epsilon}_{\boldsymbol{\theta}}$ as per-point difference between $\boldsymbol{x}^t$ and $\boldsymbol{x}^{t-1}$ (with some arithmetic ignored). In addition, $\boldsymbol{\epsilon}_{\boldsymbol{\theta}}$ should also effectively incorporate multi-level information from $\boldsymbol{c}$. The goal is to infer not only the overall shape but also the fine-grained details based on $\boldsymbol{c}$. We design a neural network that achieves these features. The overall architecture is shown in Figure 2. It is composed of two parallel sub-networks similar to PointNet++ (Qi et al., 2017b), and they have the same hierarchical structure.

The upper subnet, which we refer as the Condition Feature Extraction subnet, extracts multi-level features from the incomplete point cloud $\boldsymbol{c}$. The lower subnet, which we refer as the Denoise subnet, takes the noisy point cloud $\boldsymbol{x}^t$ as input. We also add the diffusion step $t$, the global feature extracted from $\boldsymbol{c}$, and multi-level features extracted by the Condition Feature Extraction subnet to the Denoise subnet. The diffusion step $t$ is first transformed into a 512-dimension step embedding vector through positional encoding and fully connected (FC) layers (See Appendix Section A.1 for details), and then inserted to every level of the Denoise subnet. Similarly, the conditioner $\boldsymbol{c}$ is first transformed into a 1024-length global feature through a two-stage PointNet, and then inserted to every level of the Denoise subnet. The multi-level features extracted by the Condition Feature Extraction subnet are

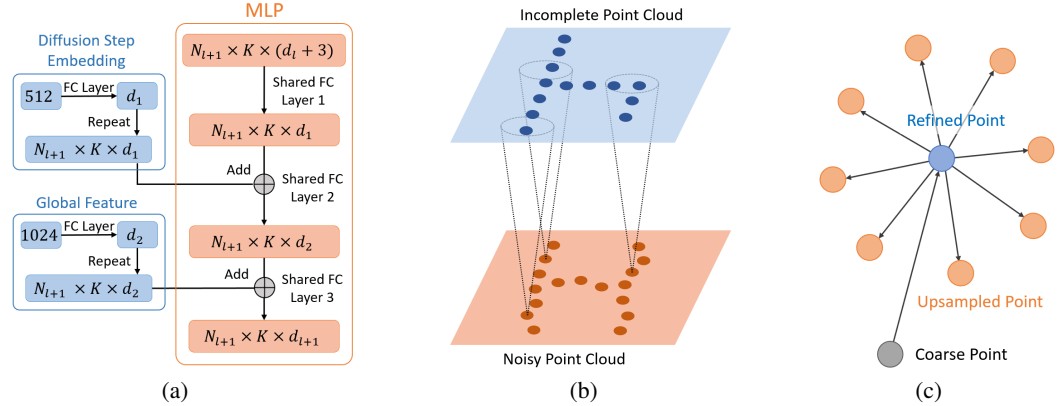

Figure 3: (a) Insert information of the diffusion step embedding and the global feature to the shared MLP. (b) The Feature Transfer module maps features from the incomplete point cloud to the noisy point cloud. (c) Refine and upsample the coarse points at the same time.

inserted to every level of the Denoise subnet through Feature Transfer modules. Finally, the Denoise subnet is connected to a shared MLP and outputs $\epsilon_{\theta}(\boldsymbol{x}^t, \boldsymbol{c}, t)$.

Additionally, while Zhou et al. (2021) argues PointNet++ cannot be used in a DDPM that generates point clouds, we find attaching the absolute position of each point to its feature solves this problem. See Appendix Section A.3 for detailed analysis. We also improve the backbone PointNet++ so that it manipulates positions of points more accurately.

In the next paragraphs, we elaborate on the building blocks of the improved PointNet++: Set Abstraction modules in the encoder, and Feature Propagation modules in the decoder, and Feature Transfer modules between the Condition Feature Extraction subnet and the Denoise subnet.

**Set Abstraction (SA) Module.** Similar to PointNet++, this module subsamples the input point cloud and propagates the input features. Assume the input is $\{x_j | 1 \leq j \leq N_l\}$, where $x_j$ is the 3D coordinate of the $j$-th point and $N_l$ is the number of input points to the Set Abstraction module of level $l$. Each point has a feature of dimension $d_l$. We concatenate these features with their corresponding 3D coordinates and group them together to form a matrix $\boldsymbol{F}_l$ of shape $N_l \times (d_l + 3)$. The SA module first uses iterative farthest point sampling (FPS) to subsample the input points to $N_{l+1}$ points: $\{y_k | 1 \leq k \leq N_{l+1}\}$. Then it finds $K$ neighbors in the input $\{x_j | 1 \leq j \leq N_l\}$ for each $y_k$. We denote the $K$ neighbors of $y_k$ as $\{x_j | j \in \mathcal{B}_x(y_k)\}$, where $\mathcal{B}_x(y_k)$ is the index set of the $K$ neighbors. See definition of neighbors in Appendix A.2. These neighbors and their features are grouped together to form a matrix $\boldsymbol{G}_{in}$ of shape $N_{l+1} \times K \times (d_l + 3)$. Then a shared multi-layer perceptron (MLP) is applied to transform the grouped feature $\boldsymbol{G}_{in}$ to $\boldsymbol{G}_{out}$, which is of shape $N_{l+1} \times K \times d_{l+1}$ and $d_{l+1}$ is the dimension of the output feature. Finally, a max-pooling is applied to aggregate features from the $K$ neighbors $\{x_j | j \in \mathcal{B}_x(y_k)\}$ to $y_k$. We obtain the output of the SA module, the matrix $\boldsymbol{F}_{l+1}$, which is of shape $N_{l+1} \times d_{l+1}$.

Note that we need to incorporate information of the diffusion step embedding and global feature extracted from the incomplete point cloud $\boldsymbol{c}$ to every SA module in the Denoise subnet as shown in Figure 2. We insert these information to the shared MLP that transforms $\boldsymbol{G}_{in}$ to $\boldsymbol{G}_{out}$ mentioned in the above paragraph. Specifically, we add them to the channel dimension of the intermediate feature maps in the shared MLP. Figure 3(a) illustrates this process in details. Inspired by the works (Pan et al., 2021; Zhao et al., 2020), we also replace the max-pooling layer in the SA module with a self-attention layer. Feature at $y_k$ is obtained by a weighted sum of the features of its $K$ neighbors $\{x_j | j \in \mathcal{B}_x(y_k)\}$ instead of max-pooling, and the weights are adaptively computed through the attention mechanism. See Appendix A.4 for details of this attention layer.

**Feature Propagation (FP) Module.** Similar to PointNet++, this module upsamples the input point cloud and propagates the input features. In PointNet++, the features are upsampled from $\{y_k | 1 \leq k \leq N_{l+1}\}$ to $\{x_j | 1 \leq j \leq N_l\}$ by three interpolation: Feature at $x_j$ is a weighted sum of the features of its three nearest neighbors in $\{y_k | 1 \leq k \leq N_{l+1}\}$. We think that the three interpolation operation is not suitable in our task, because the interpolation operation may lose some information about the accurate positions of the points. See a detailed analysis in Appendix Section A.5.

We propose to use a Point Adaptive Deconvolution (PA-Deconv) module to upsample the point features. In the SA module, the features are mapped from set $\{x_j | 1 \leq j \leq N_l\}$ to $\{y_k | 1 \leq k \leq N_{l+1}\}$. The key step is to find the neighbors $\{x_j | j \in \mathcal{B}_x(y_k)\} \subseteq \{x_j | 1 \leq j \leq N_l\}$ for each $y_k$. Features at $\{x_j | j \in \mathcal{B}_x(y_k)\}$ are transformed and then aggregated to the point $y_k$ through either max-pooling or attention mechanism. Now in the FP module, we need to map features the other way around: from $\{y_k | 1 \leq k \leq N_{l+1}\}$ to $\{x_j | 1 \leq j \leq N_l\}$. We can achieve this goal through a similar method. We find the neighbors $\{y_k | k \in \mathcal{B}_y(x_j)\} \subseteq \{y_k | 1 \leq k \leq N_{l+1}\}$ for each $x_j$. Features at $\{y_k | k \in \mathcal{B}_y(x_j)\}$ are transformed through a shared MLP, and then aggregated to the point $x_j$ through attention mechanism. Similar to SA modules, we insert information of the diffusion step embedding and the global feature extracted from the incomplete point cloud $c$ to the shared MLP in every FP module in the Denoise subnet. Finally, same as the original FP module in PointNet++, the upsampled features are concatenated with skip linked point features from the corresponding SA module, and then passed through a unit PointNet. The Feature Propagation module are applied four times and features are eventually propagated to the original input point cloud.

**Feature Transfer (FT) Module.** The FT module transmits information from the Condition Feature Extraction subnet to the Denoise subnet. Assume the point cloud at level $l$ in the Condition Feature Extraction subnet is $\{z_l | 1 \leq l \leq S_l\}$, where $S_l$ is the number of points at level $l$ in the Condition Feature Extraction subnet. The FT module maps the features at points $\{z_r | 1 \leq r \leq S_l\}$ to points at the same level in the Denoise subnet, which are $\{x_j | 1 \leq j \leq N_l\}$. Then the mapped features are concatenated with the original features at $\{x_j | 1 \leq j \leq N_l\}$. Next, the concatenated features are fed to the next level of the Denoise subnet. In this way, the Denoise subnet can utilize local features at different levels of the incomplete point cloud to manipulate the noisy input point cloud to form a clean and complete point cloud. The key step in this process is to map features at $\{z_r | 1 \leq r \leq S_l\}$ to $\{x_j | 1 \leq j \leq N_l\}$. We adopt a similar strategy in the SA module. We find the neighbors $\{z_r | r \in \mathcal{B}_z(x_j)\} \subseteq \{z_r | 1 \leq r \leq S_l\}$ for each $x_j$. Features at $\{z_r | r \in \mathcal{B}_z(x_j)\}$ are transformed through a shared MLP, and then aggregated to the point $x_j$ through the attention mechanism, which is a weighted sum of the features at $\{z_r | r \in \mathcal{B}_z(x_j)\}$.

We set a small distance to define neighbors in low level FT modules, so that they only query the adjacent parts of the incomplete point cloud $c$ to preserve local details in it. Large distances are set to define neighbors in high level FT modules. This makes high-level FT modules have large receptive fields, so that they can query a large part of the incomplete point cloud to infer high level 3D structural relations. See detailed neighbor definitions in Appendix Section A.2.

## 3.3 REFINEMENT NETWORK

We denote the coarse point cloud generated by the Conditional Generation Network as $U$. We use another network of the same architecture shown in Figure 2 to predict a per-point displacement for $U$ to refine it. The differences are that the input to the Denoise subnet becomes $U$ and we do not need to insert the diffusion step embedding to the Denoise subnet. The predicted displacement are added to $U$ to obtain the refined point cloud $V$: $v = u + \gamma \epsilon_f(u, c)$, where $v, u, c$ are the concatenated 3D coordinates of the point clouds $V, U, C$, respectively. $\gamma$ is a small constant and we set it to 0.001 in all our experiments. $\epsilon_f$ is the ReFinement Network. We use the Chamfer Distance (CD) loss between the refined point cloud $V$ and ground truth point cloud $X$ to supervise the network $\epsilon_f$:

$$\mathcal{L}_{\text{CD}}(V, X) = \frac{1}{|V|} \sum_{v \in V} \min_{x \in X} ||v - x||^2 + \frac{1}{|X|} \sum_{x \in X} \min_{v \in V} ||v - x||^2, \tag{5}$$

where $|V|$ means number of points in $V$. If we also want to upsample points in $U$ by a factor of $\lambda$, we can simply increase the output dimension of the network $\epsilon_f$. In addition to predicting one 3D displacement of each point in $U$, we predict another $\lambda$ displacements. We consider each point in the refined point cloud $V$ as the center of a group of $\lambda$ points in the dense point cloud that we want to generate. The additional $\lambda$ displacements are added to every point in $V$ to form a dense point cloud. Figure 3(c) illustrates how we upsample every point in $V$ by a factor of $\lambda = 8$.

When training the ReFinement Network $\epsilon_f$, parameters in the Conditional Generation Network $\epsilon_\theta$ are fixed. It is not practical to generate coarse point clouds $U$ on the fly in the training process of $\epsilon_f$, because the generation process of DDPM is slow. Instead, we generate and save the coarse point clouds in advance. Due to the probabilistic nature of DDPM, we generate 10 coarse point clouds for each incomplete point cloud in the dataset to increase diversity of training data.

# 4 RELATED WORKS

**Point cloud completion.** Inspired by the pioneering work, PointNet (Qi et al., 2017a), researchers focus on learning global feature embeddings from 3D point clouds for completion (Yuan et al., 2018; Tchapmi et al., 2019), which however cannot predict local and thin shape structures. To address these challenges, following research works (Pan, 2020; Xie et al., 2020; Zhang et al., 2020; Wen et al., 2021; Yu et al., 2021; Pan et al., 2021) exploit multi-scale local point features to reconstruct complete point clouds with fine-grained geometric details. Recently, PointTr (Yu et al., 2021) and VRCNet (Pan et al., 2021) provide impressive point cloud completion results with the help of attention-based operations. Nonetheless, as a challenging conditional generation problem, point cloud completion has not been fully resolved.

**DDPM for point cloud generation.** Luo & Hu (2021) are the first to use DDPM for unconditional point cloud generation. They use a Pointwise-net to generate point clouds, which is similar to a 2-stage PointNet used for point cloud part segmentation. However, the Pointwise-net could only receive a global feature. It can not leverage fine-grained local structures in the incomplete point cloud. Zhou et al. (2021) further use conditional DDPM for point cloud completion by training a point-voxel CNN (Liu et al., 2019), but the way they use the incomplete point cloud $c$ is different from ours. They directly concatenate $c$ with the noisy input $x^t$, and feed them to a single point-voxel CNN. This may hurt performance of the network, because the concatenated point cloud is very likely to be non-uniform. In addition, $x^t$ is very different from $c$ for large $t$'s due to the large noise magnitude in $x^t$. Feeding two point clouds of very different properties to a single network at once could be quite confusing for the network. The other major difference is that they do not refine or upsample the coarse point cloud generated by DDPM like we do.

# 5 EXPERIMENTS

## 5.1 DATASETS

We conduct point cloud completion experiments on the following three datasets. **MVP.** The MVP dataset (Pan et al., 2021) has 62400 training partial-complete point cloud pairs and 41600 testing pairs sampled from ShapeNet (Chang et al., 2015). Every partial point cloud has 2048 points. In particular, MVP dataset provides ground truth point clouds with different resolutions, including 2048, 4096, 8192, and 16384 points. **MVP-40.** The MVP-40 dataset (Pan et al., 2021) consists of 41600 training samples and 64168 testing samples from 40 categories in ModelNet40 (Wu et al., 2015). Its partial point clouds are sampled from complete point clouds with a pre-defined missing ratio, *i.e.*, 50%, 25% and 12.5% missing. Both the partial and complete point clouds have 2048 points. **Completion3D.** It (Tchapmi et al., 2019) consists of 28974 point cloud pairs for training and 1184 for testing from 8 object categories in ShapeNet. Both the partial and complete point clouds have 2048 points. We find some pairs of the incomplete point cloud and complete point cloud have inconsistent scales in the Completion3D dataset. We correct the scales and use the corrected dataset in our experiments. See details in Appendix Section B.4.

## 5.2 EVALUATION METRICS

We use the Chamfer Distance (CD), Earth Mover Distance (EMD), and F1 score to evaluate the quality of the generated point clouds. CD distance is defined in Equation 5.

**Earth Mover Distance.** Consider the predicted point cloud $V$ and the ground truth point cloud $X$ of equal size $N = |V| = |X|$, the EMD loss penalizes their shape discrepancy by optimizing a transportation problem. It estimates a bijection $\phi : V \longleftrightarrow X$ between $V$ and $X$:

$$\mathcal{L}_{\text{EMD}}(V, X) = \min_{\phi : V \longleftrightarrow X} \sum_{v \in V} \|v - \phi(v)\|_2.$$

(6)

**F1 score.** To compensate the problem that CD loss can be sensitive to outliers, we follow previous methods (Pan et al., 2021; Tatarchenko et al., 2019) and use F1 score to explicitly evaluates the distance between object surfaces, which is defined as the harmonic mean between precision $\mathcal{L}_{\text{P}}(\rho)$ and recall $\mathcal{L}_{\text{R}}(\rho)$: $\mathcal{L}_{\text{F1}} = \frac{2\mathcal{L}_{\text{P}}(\rho)\mathcal{L}_{\text{R}}(\rho)}{\mathcal{L}_{\text{P}}(\rho) + \mathcal{L}_{\text{R}}(\rho)}$, where $\mathcal{L}_{\text{P}}(\rho) = \frac{1}{|V|} \sum_{v \in V} \left[ \min_{x \in X} \|x - v\|^2 < \rho \right]$, $\mathcal{L}_{\text{R}}(\rho) = \frac{1}{|X|} \sum_{x \in X} \left[ \min_{v \in V} \|x - v\|^2 < \rho \right]$, and $\rho$ is a predefined distance threshold. We set $\rho = 10^{-4}$ for the MVP and Completion3D datasets, and set $\rho = 10^{-3}$ for the MVP-40 dataset.

Table 1: Point cloud completion results on MVP, MVP-40 and Completion3D datasets at the resolution of 2048 points. CD loss is multiplied by $10^4$. EMD loss is multiplied by $10^2$. Scale factors of the two losses are the same in all the other tables. The two losses are the lower the better, while F1 score is the higher the better. Note that MVP-40 dataset has larger CD and EMD losses because objects in it have larger scales than the other two datasets. Results of MVP-40 dataset at 25% missing ratio is complemented in Appendix Table 5.

| Method | MVP | | | MVP40 (50% missing) | | | MVP40 (12.5% missing) | | | Completion3D | | |
|---|---|---|---|---|---|---|---|---|---|---|---|---|
| | CD | EMD | F1 | CD | EMD | F1 | CD | EMD | F1 | CD | EMD | F1 |
| PCN (Yuan et al., 2018) | 8.65 | 1.95 | 0.342 | 39.67 | 6.37 | 0.581 | 32.56 | 6.18 | 0.619 | 8.81 | 3.03 | 0.315 |
| TopNet (Tchapmi et al., 2019) | 10.19 | 2.44 | 0.299 | 48.52 | 8.75 | 0.506 | 40.12 | 9.08 | 0.542 | 11.56 | 3.69 | 0.257 |
| FoldingNet (Yang et al., 2018) | 10.54 | 3.64 | 0.256 | 51.89 | 11.66 | 0.441 | 46.03 | 8.93 | 0.480 | 14.32 | 4.81 | 0.186 |
| MSN (Liu et al., 2020) | 7.08 | 1.71 | 0.434 | 34.33 | 9.70 | 0.646 | 20.20 | 4.54 | 0.728 | 8.88 | 2.69 | 0.359 |
| Cascade (Wang et al., 2020) | 6.83 | 2.14 | 0.436 | 34.16 | 15.40 | 0.635 | 26.73 | 5.71 | 0.657 | 7.31 | 2.70 | 0.408 |
| ECG (Pan, 2020) | 7.06 | 2.36 | 0.443 | 34.06 | 16.19 | 0.671 | 40.00 | 6.98 | 0.597 | 10.43 | 3.63 | 0.300 |
| GRNet (Xie et al., 2020) | 7.61 | 2.36 | 0.353 | 35.99 | 12.33 | 0.589 | 22.04 | 6.43 | 0.646 | 8.54 | 2.87 | 0.314 |
| PMPNet (Wen et al., 2021) | 5.85 | 3.42 | 0.475 | **25.41** | 29.92 | 0.721 | 13.00 | 8.92 | 0.815 | 7.45 | 4.85 | 0.386 |
| VRCNet (Pan et al., 2021) | 5.82 | 2.31 | 0.495 | 25.70 | 18.40 | 0.736 | 14.20 | 5.90 | 0.807 | **6.69** | 3.57 | 0.433 |
| PDR paradigm (Ours) | **5.66** | **1.37** | **0.499** | 27.20 | **2.68** | **0.739** | **12.70** | **1.39** | **0.827** | 7.10 | **1.75** | **0.451** |

Table 2: Completion results on MVP dataset at the resolution of 4096, 8192, 16384 points.

| # Points | 4096 | | 8192 | | 16384 | |
|---|---|---|---|---|---|---|
| | CD | F1 | CD | F1 | CD | F1 |
| PCN | 7.14 | 0.469 | 6.02 | 0.577 | 5.18 | 0.650 |
| TopNet | 7.69 | 0.434 | 6.64 | 0.526 | 5.14 | 0.618 |
| FoldingNet | 8.76 | 0.351 | 6.90 | 0.433 | 6.98 | 0.464 |
| MSN | 5.37 | 0.583 | 4.40 | 0.663 | 4.09 | 0.696 |
| Cascade | 5.46 | 0.579 | 4.51 | 0.686 | 3.90 | 0.743 |
| ECG | 7.31 | 0.506 | 3.99 | 0.717 | 3.32 | 0.774 |
| GRNet | 5.73 | 0.493 | 4.51 | 0.616 | 3.54 | 0.700 |
| PoinTr | 4.29 | 0.638 | 3.52 | 0.725 | 2.95 | 0.783 |
| VRCNet | 4.62 | 0.629 | 3.39 | 0.734 | 2.81 | 0.780 |
| Ours | **4.26** | **0.649** | **3.35** | **0.754** | **2.61** | **0.817** |

Table 3: Comparison of different network structures in term of training the conditional generation network and refinement network.

| Task | Network | CD | EMD | F1 |
|---|---|---|---|---|
| Generate Coarse Points | Pointwise-net | 11.99 | 1.63 | 0.265 |
| | Concate $x^t$ & $c$ | 10.79 | 1.54 | 0.382 |
| | PointNet++ | 9.39 | 1.38 | 0.355 |
| | PA-Deonv | 8.81 | 1.34 | 0.379 |
| | PA-Deonv & Att. | **8.71** | **1.29** | **0.389** |
| Refine Coarse Points | Pointwise-net | 7.71 | 1.45 | 0.407 |
| | Concate $x^t$ & $c$ | 5.78 | 1.38 | 0.490 |
| | PointNet++ | 6.03 | 1.40 | 0.480 |
| | PA-Deonv | 5.96 | 1.40 | 0.482 |
| | PA-Deonv & Att. | **5.66** | **1.37** | **0.499** |

## 5.3 POINT CLOUD COMPLETION

We compare our point cloud completion method with previous state-of-the-art point cloud completion methods. The comparison is performed on MVP, MVP-40, and Completion3D datasets. Results are shown in Table 1. We also conduct multi-resolution experiments on the MVP dataset, and results are shown in Table 2. Detailed experimental setups are provided in Appendix Section B.1. We can see that our Conditional **P**oint **D**iffusion-**R**efinement (PDR) paradigm outperforms other methods by a large margin in terms of EMD loss, which is highly indicative of uniformness (Zhang et al., 2021). We also achieve the highest F1 score and very low CD loss. Although VRCNet sometimes has lower CD loss than ours, it tends to put more points in the parts that are known in the incomplete point clouds, while put less points in the missing part (See Figure 10 in Appendix). In this way, its CD loss could be very low, but this non-uniformness is undesired and leads to very high EMD loss. We compare our method with other baselines in terms of visual quality of completed point clouds in Figure 4. We can see that our method generally has better visual quality. More samples are provided in Figure 9 and Figure 11 in Appendix. We also find that our PDR paradigm demonstrate some diversity in completion results as discussed in Appendix B.8.

**Ablation Study.** We study the effect of attention mechanism, Point Adaptive Deconvolution (PA-Deconv) module, and Feature Transfer (FT) module in term of training the Conditional Generation Network and the Refinement Network. The experiments are conducted on MVP dataset at the resolution of 2048 points and results are shown in Table 3. "PA-Deonv & Att." is our proposed complete network shown in Figure 2. "PA-Deonv" removes attention mechanism. "PointNet++" further removes PA-Deconv module. "Concate $x^t$ & $c$" removes FT modules. It concatenates $c$ with $x^t$ as Zhou et al. (2021) do, and feed them to a single PointNet++ with attention mechanism and PA-Deonv. "Pointwise-net" only utilizes a global feature extracted from the incomplete point cloud. We can see that these proposed modules indeed improve the networks' performance. Note that the conditional generation networks in Table 3 are trained without data augmentation. Complete experimental results with data augmentation are presented in Appendix Section B.6. All the refinement networks are trained using data generated by our proposed complete dual-path network trained with data augmentation. If the other ablated networks use training data generated by themselves, they would have worse refinement results.

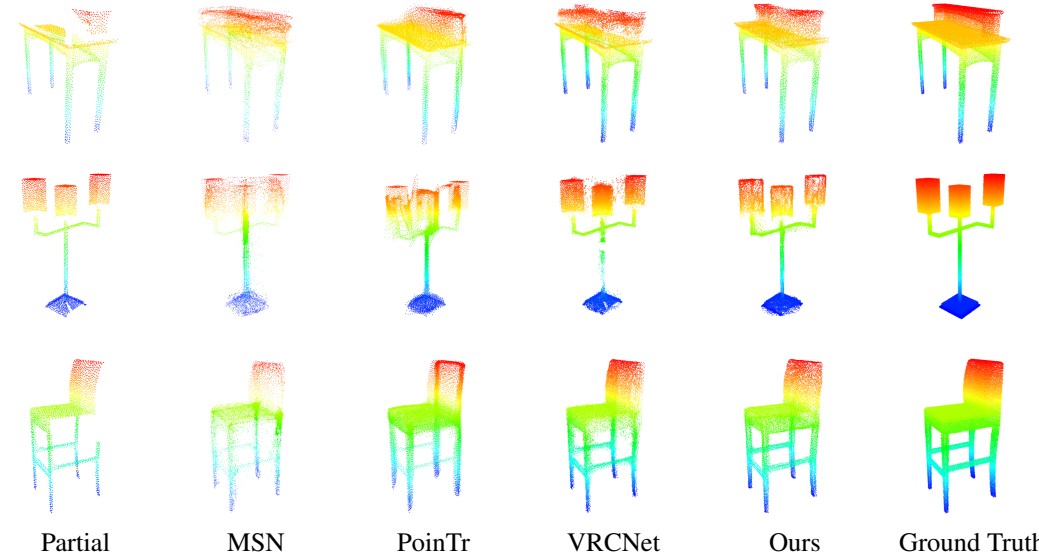

|  Partial | MSN | PoinTr | VRCNet | Ours | Ground Truth |

Figure 4: Visual comparison of point cloud completion results on the MVP dataset (16384 points).

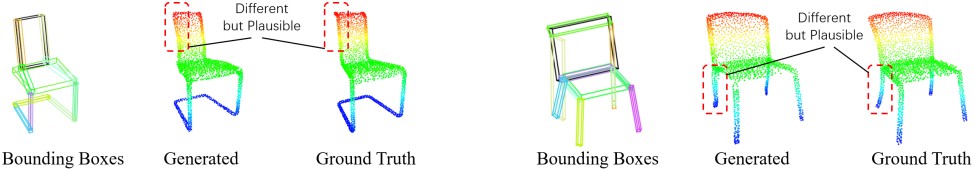

| Bounding Boxes | Generated | Ground Truth | | Bounding Boxes | Generated | Ground Truth |

Figure 5: Our method can be extended to controllable point cloud generation.

**DDPM acceleration.** Kong & Ping (2021) propose to accelerate the generation process of DDPM by jumping steps in the reverse process. The method does not need retraining of the DDPM. We directly apply their method to our 3D point cloud generation network. However, we observe a considerable performance drop in the accelerated DDPM. On the MVP dataset (2048 points), the original 1000-step DDPM achieves $10.7 \times 10^{-4}$ CD loss. CD losses of the accelerated 50-step and 20-step DDPMs increase to $13.2 \times 10^{-4}$ and $18.1 \times 10^{-4}$, respectively. Fortunately, we can generate coarse point clouds using the accelerated DDPMs and use another Refinement Network to refine them. The refined point clouds of the 50-step and 20-step DDPMs bear CD losses of $5.68 \times 10^{-4}$ and $5.78 \times 10^{-4}$, respectively. Compared with the original 1000-step DDPM, which has a CD loss of $5.66 \times 10^{-4}$, it's quite temping to accept a slight drop in performance for an acceleration up to 50 times. Complete results of the acceleration experiment are presented in Appendix Section B.7.

### 5.4 EXTENSION TO CONTROLLABLE GENERATION

Our conditional PDR paradigm can be readily extended to controllable point cloud generation conditioned on bounding boxes of every part of an object. We sample points on the surfaces of the bounding boxes and regard this point cloud as the conditioner, just like the incomplete point cloud serves as the conditioner for point cloud completion. We conduct experiments on the chair category of PartNet (Mo et al., 2019) dataset. Two examples are shown in Figure 5. It's interesting that our method can generate a shape different from the ground truth in some details, but be still plausible.

## 6 CONCLUSION

In this paper, we propose the Conditional **P**oint **D**iffusion-**R**efinement (PDR) paradigm for point cloud completion. Our method effectively leverages the strong spatial correspondence between the adjacent parts of the incomplete point cloud and the complete point cloud through the proposed Feature Transfer module, which could also infer high-level 3D structural relations. We make improvements of the backbone PointNet++ to make it capable of accurately manipulating positions of input points. Our method demonstrate significant advantages over previous methods, especially in terms of the overall distribution of the generated point cloud. We also find that our method has great potential to be applied in other conditional point cloud generation tasks such as controllable point cloud generation.

## 7 ACKNOWLEDGEMENTS

This work is partially supported by General Research Fund (GRF) of Hong Kong (No. 14205719). The authors thank useful discussions with Quan Wang from SenseTime and Tong Wu from CUHK.

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

APPENDIX

# A  METHODOLOGY DETAILS

## A.1  DETAILS OF THE DDPM

**Hyperparameters $\beta_t$.**  We define hyperparameters $\beta_t$ in the diffusion process according to a linear schedule. We let $\beta_1 = 1 \times 10^{-4}$ and $\beta_T = 2 \times 10^{-2}$. Then, we define $\beta_t = \frac{t-1}{T-1} \cdot (\beta_T - \beta_1), t = 1, 2, \cdots, T$.

**Diffusion step embedding.**  The network $\epsilon$ needs to output different $\epsilon_{\boldsymbol{\theta}}(\boldsymbol{x}^t, \boldsymbol{c}, t)$ for different diffusion steps $t \in \{1, \cdots, T\}$. We first use positional encoding (Vaswani et al., 2017) to encode each $t$ into a $2d_t$ dimensional vector $\boldsymbol{\phi}_{\mathrm{emb}}(t) = [\sin(\boldsymbol{\psi}(t)), \cos(\boldsymbol{\psi}(t))]$, where

$$\boldsymbol{\psi}(t) = \left[ 10^{\frac{4 \times 0}{d_t} t}, 10^{\frac{4 \times 1}{d_t} t}, \cdots, 10^{\frac{4 \times (d_t - 1)}{d_t} t} \right].$$

We set $d_t = 64$ in experiments. Then, we use two fully-connected (FC) layers to transform $\boldsymbol{\phi}_{\mathrm{emb}}(t)$ into a 512 dimensional embedding vector (Ho et al., 2020). The first FC layer has input dimension $2d_t = 128$ and output dimension 512. The second FC layer has input dimension 512 and output dimension 512. Both layers are followed by the Swish activation function (Ramachandran et al., 2017).

## A.2  NEIGHBOR DEFINITION

In the Set Abstraction module and the Feature Transfer module, the neighbors are defined as the points that are within a specified distance to the center point. If a center point has more than $K$ neighbors, we randomly select $K$ neighbors from its neighbors. If a center point has less than $K$ neighbors, we pad it with dummy neighbors that has the same position as the center point, and has features of zeros. These dummy neighbors are excluded from the max-pooling operation. And in the attention mechanism, the weights of the dummy neighbors are manually set to 0. In this way, we can guarantee that a center point can always find $K$ neighbors. We set $K = 32$ in the Set Abstraction module and the Feature Transfer module. In the 4 levels of the Set Abstraction modules, the neighboring distance are set to $0.1, 0.2, 0.4, 0.8$, respectively. In the 9 Feature Transfer modules, the neighboring distance are set to $0.1, 0.2, 0.4, 0.8, 1, 6, 0.8, 0.4, 0.2, 0.1$, respectively. Coordinates of samples in all datasets are normalized to the range $[-1, 1]$.

In the Point Adaptive Deconvolution (PA-Deconv) modules of the Feature Propagation modules, the $K$ neighbors are defined as the $K$ nearest neighbors of the center point, and we set $K = 8$ for all Feature Propagation modules.

## A.3  PROBLEMS OF VANILLA POINTNET++

Zhou et al. (2021) argue that PointNet++ can not be used to train a DDPM. We observe the same phenomenon in our experiments. We find that this is because the density of the input noisy cloud $\boldsymbol{x}^t$ is too low for large $t$'s. Recall that $\boldsymbol{x}^t = \sqrt{\bar{\alpha}_t} \boldsymbol{x}^0 + \sqrt{1 - \bar{\alpha}_t} \boldsymbol{\epsilon}$, where $\boldsymbol{\epsilon}$ is a Gaussian noise. $\bar{\alpha}_t$ goes to 0 for large $t$'s. This means $\boldsymbol{x}^t$ is close to a Gaussian noise when $t$ is large. The density of a Gaussian noise is much lower than the complete point cloud $\boldsymbol{x}^0$. This is because points in $\boldsymbol{x}^0$ concentrate on the surface of some object and $\boldsymbol{x}^0$ is normalized to the range $[-1, 1]$, while points from a standard Gaussian distribution could fill the whole space in the range of $[-3, 3]$.

PointNet++ is originally designed to process point clouds like $\boldsymbol{x}^0$. It's selection of distances to define neighbors described in Appendix Section A.2 is suitable for point clouds that have the same level of density as $\boldsymbol{x}^0$, but can not handle point clouds close to a Gaussian noise. Indeed, we find the average number of neighbors for each point in the four levels of the Set Abstraction modules are $22.3984, 29.9133, 29.3266, 27.8375$, respectively, for 10 random shapes sampled from the MVP dataset. In constrast, the average number of neighbors for each point in the four levels are $1.0864, 1, 1, 1$ for 10 random point clouds sampled from the Gaussian distribution. Note that each point itself is considered to be a neighbor of itself. This means most points do not have any neighbors besides itself in a Gaussian noise.

PointNet++ only utilizes the relative positions of input points. The input feature of each point to the first Set Abstraction module is its relative position to the center point, which is subsampled from the original input points by farthest point sampling. No information can be extracted when points do not have neighbors. This is the reason why PointNet++ can not be directly used to train a DDPM.

Our solution is to attach the absolute position of each point to its feature. This guarantees that a point at least can utilize its own position to decide which direction to move when it does not have neighbors. Afterall, a point cloud with large magnitude noises does not have many meaningful structures. There is not much information in the relative positions of points. Another solution is to change the definition of neighbors: From points within a specified distance to K-nearest neighbors. This guarantees that a point always has $K$ neighbors. We conduct experiments to compare these two solutions, and we find that their performances are basically the same. Therefore, we just stick to the original neighbor definitions in PointNet++.

### A.4 ATTENTION MECHANISM

In Section 3.2 in the main text, we mentioned that we use the attention mechanism instead of max-pooling to aggregate features at the neighboring points to the center point. We take the Set Abstraction module as an example to elaborate on the attention mechanism. Attention mechanism in the Feature Propagation module and Feature Transfer module is similarly designed.

Assume we want to propagate features from $\{x_j | 1 \leq j \leq N_l\}$ to $\{y_k | 1 \leq k \leq N_{l+1}\}$. Each point in $\{x_j | 1 \leq j \leq N_l\}$ has a feature of dimension $d_l$. We concatenate these features with their corresponding 3D coordinates and group them together to form a matrix $\boldsymbol{F}_l$ of shape $N_l \times (d_l + 3)$. We finds $K$ neighbors in the input set $\{x_j | 1 \leq j \leq N_l\}$ for each $y_k$. These neighbors together with their features are grouped together to form a matrix $\boldsymbol{G}_{in}$ of shape $N_{l+1} \times K \times (d_l + 3)$. Then a shared multi-layer perceptron (MLP) is applied to transform the grouped feature $\boldsymbol{G}_{in}$ to $\boldsymbol{G}_{out}$, which is a matrix of shape $N_{l+1} \times K \times d_{l+1}$ and $d_{l+1}$ is the dimension of the output feature.

In our attention mechanism, $\boldsymbol{G}_{in}$ (shape $N_{l+1} \times K \times (d_l + 3)$) will act like keys, $\boldsymbol{G}_{out}$ (shape $N_{l+1} \times K \times d_{l+1}$) will act like values, while the original features at $\{y_k | 1 \leq k \leq N_{l+1}\}$ will act like queries. Since $\{y_k | 1 \leq k \leq N_{l+1}\}$ is a subset of $\{x_j | 1 \leq j \leq N_l\}$, we can group the original features at $\{y_k | 1 \leq k \leq N_{l+1}\}$ to form a matrix $\boldsymbol{Q}$, which is of shape $N_{l+1} \times (d_l + 3)$. $\boldsymbol{Q}$ is first repeated $K$ times into a matrix of shape $N_{l+1} \times K \times (d_l + 3)$. Then this matrix is passed through a shared MLP and transformed into a matrix $\boldsymbol{Q}'$, which is of shape $N_{l+1} \times K \times d_{query}$. Next, we pass $\boldsymbol{G}_{in}$ through a shared MLP to transform it into a new matrix $\boldsymbol{G}'_{in}$, which is of shape $N_{l+1} \times K \times d_{key}$. We concatenate the query matrix $\boldsymbol{Q}'$ with the key matrix $\boldsymbol{G}'_{in}$ along the feature dimension. We denote this matrix as $[\boldsymbol{Q}', \boldsymbol{G}'_{in}]$, which is of shape $N_{l+1} \times K \times (d_{query} + d_{key})$. $[\boldsymbol{Q}', \boldsymbol{G}'_{in}]$ is passed through a shared MLP to obtain the scores of all the $K$ neighbors. We denote the scores as matrix $\boldsymbol{S}$ of shape $N_{l+1} \times K \times d_{l+1}$. Note that $\boldsymbol{S}$ has the same shape as $\boldsymbol{G}_{out}$. And the scores $\boldsymbol{S}$ of the $K$ neighbors are adaptively computed according to the feature at the center point $y_k$ and features at its $K$ neighbors $\{x_j | j \in \mathcal{B}_x(y_k)\}$. We apply a softmax operation along the neighbor dimension (the second dimension) of $\boldsymbol{S}$ to obtain the weight matrix of all the $K$ neighbors. We denote it as $\boldsymbol{W}$, which is of shape $N_{l+1} \times K \times d_{l+1}$. Note that we manually set the weights of the padded dummy neighbors to 0 in $\boldsymbol{W}$. Then the weight matrix $\boldsymbol{W}$ and the value matrix $\boldsymbol{G}_{out}$ are dot producted along the neighbor dimension (the second dimension) to form the output matrix $\boldsymbol{F}'$, which is of shape $N_{l+1} \times d_{l+1}$. Finally, $\boldsymbol{F}'$ is concatenated with the 3D coordinates of the set $\{y_k | 1 \leq k \leq N_{l+1}\}$ to form output of the Set Abstraction module, $\boldsymbol{F}_{l+1}$, which is a matrix of shape $N_{l+1} \times (d_{l+1} + 3)$.

### A.5 PROBLEMS WITH THREE INTERPOLATION

The original PointNet++ uses three interpolation to upsample features in the Feature Propagation module. We think that the three interpolation operation is suitable for tasks like point cloud part segmentation, but not suitable in our task. Interpolation means that points close to each other have similar features. Points close to each other tend to have similar semantic labels in a clean point cloud, therefore it is meaningful to use interpolation operation to upsample features in the part segmentation task. However, in our task, the network need to predict a per-point displacement for all points in a noisy point cloud and move it towards a clean point cloud. Points close to each other

do not need to move in a similar direction in general. In fact, they may just need to move towards the opposite direction to form a smooth surface.

We also find that the three interpolation operation lacks the ability to manipulate positions of points accurately in small scales. We first elaborate on the three interpolation operation used in the original PointNet++. Assume we want to upsample features at $\{y_k|1 \leq k \leq N_{l+1}\}$ to $\{x_j|1 \leq j \leq N_l\}$, where $\{y_k|1 \leq k \leq N_{l+1}\}$ is a subset of $\{x_j|1 \leq j \leq N_l\}$.

For each $x_j$, assume its three nearest neighbors in $\{y_k|1 \leq k \leq N_{l+1}\}$ are $\{y_k|k \in \mathcal{B}_{y,3}(x_j)\}$. Then feature at $x_j$ is obtained through the following equation:

$$f(x_j) = \frac{\sum_{k \in \mathcal{B}_{y,3}(x_j)} w(y_k, x_j) f(y_k)}{\sum_{k \in \mathcal{B}_{y,3}(x_j)} w(y_k, x_j)}, \text{ where } w(y_k, x_j) = \frac{1}{||y_k - x_j||^2}, \tag{7}$$

$f(y_k)$ and $f(x_j)$ are features at $y_k$ and $x_j$, respectively. We can see that the value of $f(x_j)$ is determined by the relative distances between itself and its three nearest neighbors. However, in 3D space, the point that has a specific relative distances to three fixed points is not unique. In fact, the point can move freely on a curve.

Let's take a very simple example, assume the three nearest neighbors of $x_j$ forms a regular triangle. If we move $x_j$ along the straight line that passes the center of the triangle and is perpendicular to the plane determined by the triangle, then $x_j$ will always have the same relative distances from the three points, which means $x_j$ will always have the same interpolated feature, as long as its movement is small enough that its three nearest neighbors do not change.

This property makes the three interpolation operation not able to distinguish some specific points in a small scale, since these points could have the same interpolated value. Therefore, three interpolation operation is not suitable for our task, as we need to accurately manipulate positions of points to make them form a meaningful shape with smooth surfaces and sharp details.

# B  EXPERIMENT

## B.1  DETAILED EXPERIMENTAL SETUP

In all experiments, we use the Adam optimizer with a learning rate of $2 \times 10^{-4}$. For experiments of our PDR paradigm in Table 1 and Table 2 in the main text, we use data augmentation described in Appendix Section B.3. We train our Conditional Generation Network for 340 epochs, 200 epochs, and 500 epochs on the MVP, MVP-40, and Completion3D datasets, respectively. We save a checkpoint and evaluate the network's performance on both the training set and the test set every 20 epochs. Since the generation process of DDPM is very slow, we randomly select 1600 samples from the training set and test set respectively for evaluation. (Test set of the Completion3D dataset has less than 1600 samples. Therefore, we use all samples in the test set for evaluation.) The checkpoint with the lowest CD loss is chosen as the best network. It is used to generate training data for the Refinement Network. We train the Refinement Network for 100 epochs, 150 epochs and 200 epochs on the MVP, MVP-40, and Completion3D datasets, respectively.

Note that we subsampling the test set only when we try to choose a best checkpoint in the training process of the conditional generation network in DDPM. After choosing the best checkpoint, we use it to generate training data to train the refinement network. However, when we evaluate the whole PDR paradigm (composed of the conditional generation network and refinement network) and compare with previous methods, we evaluate them on the complete test set. Therefore, the comparison result in Table 1 and Table 2 in the main text is reliable and fair.

For ablation studies in Table 3, the Conditional Generation Networks are trained without data augmentation for 300 epochs. All the Refinement Networks are trained on the same data generated by our proposed Conditional Generation Network trained with data augmentation. The Refinement Networks are trained for 100 epochs.

All baseline methods are rerun under the data augmentation described in Appendix Section B.3 according to their open source codes. And CD loss is chosen to train all the baseline methods.

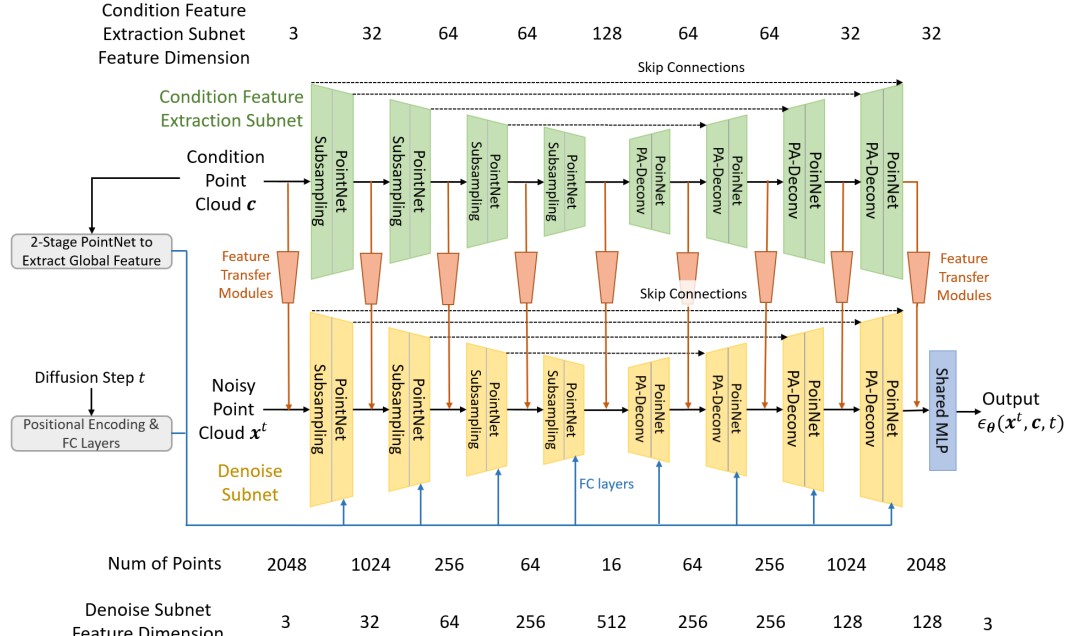

Figure 6: Detailed network structure.

## B.2 DETAILS OF THE NETWORK STRUCTURE.

Detailed network structure is shown in Figure 6. We present the number of points and feature dimension in each level of the Feature Extraction network and the Denoise network. The distances to define neighbors are provided in Appendix Section A.2.

B.3  DATA AUGMENTATION

We use rotation, mirror, translation, and scaling as data augmentation methods during training. Rotation is performed along the upward direction of the shapes. The upward direction is the $y$-axis in MVP dataset and Completion3D dataset, while upward direction in the MVP-40 dataset is the $z$-axis. And the rotation angle is uniformly sampled from the interval $[-a, a]$, where $a$ is a predefined hyper-parameter that controls the magnitude of the rotation.

Mirror operation is performed with respect to the two planes that are parallel to the upward direction: $x = 0$ plane and $z = 0$ plane for MVP dataset and Completion3D dataset, $x = 0$ plane and $y = 0$ plane for MVP-40 dataset. The mirror operation is performed with a probability of $m/2$ with respect to the two planes, respectively. $m$ is a predefined hyper-parameter the controls the probability of the mirror operation.

Same as some previous works (Wang et al., 2020; Xia et al., 2021), we observe that most objects in the MVP dataset and Completion3D dataset have reflection symmetry with respect to the $xy$ plane. Therefore, we mirror the partial input with respect to this plane and concatenate the mirrored points with the original partial input for these two datasets. We subsample this concatenated point cloud from 4096 points to 3072 points by farthest point sampling to obtain a uniform point cloud. We label the original points with 1 and the mirrored points with $-1$. This concatenated point cloud is feed to both the Conditional Generation Network and the Refinement Network, so that they could learn whether an object has reflection symmetry and determine whether to utilize the mirrored points according to their $-1$ label.

Translation is achieved by adding a randomly sampled 3D vector to every point in the incomplete point cloud and the complete point cloud. Each component of the 3D translation vector is sampled from a Gaussian distribution with zero mean and standard deviation of $\sigma$, where $\sigma$ is a predefined hyper-parameter the controls the magnitude of the translation operation.

We also randomly samples a scaling factor uniformly from the interval $[\delta_{low}, \delta_{high}]$ when loading a training pair. The scaling factor is multiplied to the coordinates of all points in the incomplete point cloud and the complete point cloud.

We observe that data augmentations can prevent the network from overfitting on the training set, but could also lead to performance drop on the test set. Therefore, we use different data augmentation schemes to train the Conditional Generation Network in the DDPM and the Refinement Network. When training the Conditional Generation Network, we hope the network does not overfit on the training set, because we need it to generate training samples to train the Refinement Network. Therefore, we use data augmentations of large magnitudes to train the Conditional Generation Network. However, when training the Refinement Network, high performance is the top priority. Therefore, we use data augmentations of small magnitudes to train the Refinement Network. We also use data augmentation to train baseline methods. The data augmentations are the same ones that we use to train the Refinement Network. The details of the data augmentation is shown in Table 4.

Table 4: Data augmentations used in MVP, MVP-40 and Completion3D dataset by the conditional generation network, refinement network, and all baselines.

| | Conditional Generation Network | | | | Refinement Network and other Baselines | | | |
|---|---|---|---|---|---|---|---|---|
| | Rotation | Mirror | Translation | Scaling | Rotation | Mirror | Translation | Scaling |
| MVP | $a = 90°$ | $m = 0.5$ | $\sigma = 0.1$ | $[1/1.2, 1.2]$ | $a = 3°$ | $m = 0.5$ | $\sigma = 0.005$ | $[1/1.01, 1.01]$ |
| MVP-40 | $a = 0°$ | $m = 0.5$ | $\sigma = 0$ | $[1/1.2, 1.2]$ | $a = 3°$ | $m = 0.5$ | $\sigma = 0.005$ | $[1/1.01, 1.01]$ |
| Completion3D | $a = 10°$ | $m = 0.2$ | $\sigma = 0.01$ | $[0.66, 1]$ | $a = 3°$ | $m = 0.1$ | $\sigma = 0.005$ | $[0.66, 1]$ |

B.4  SCALE-INCONSISTENCY ISSUE OF THE COMPLETION3D DATASET

We find that many pairs of incomplete-complete point clouds have inconsistent scales in the Completion3D dataset. A few inconsistent examples are shown in Figure 7. Ideally, the incomplete point should overlap with the complete point cloud in 3D space, but many incomplete-complete pairs in the Completion3D dataset cannot overlap with each other due to inconsistent scales, which misleads

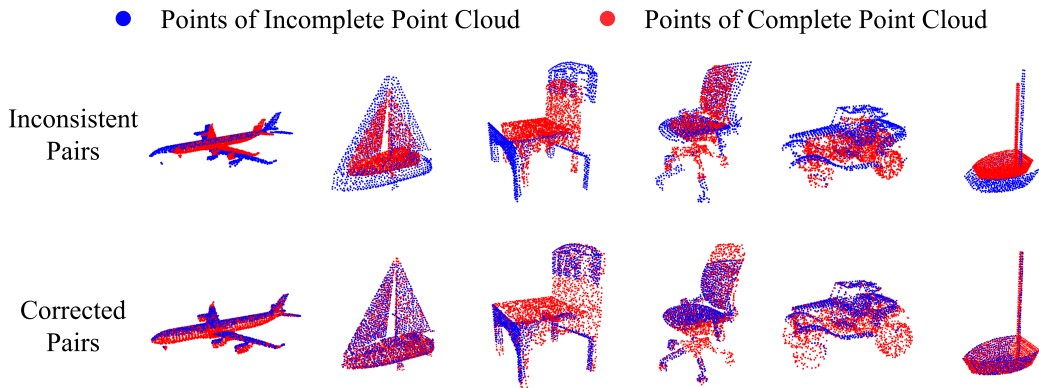

Figure 7: The first row shows some inconsistent pairs of the incomplete point cloud and complete point cloud from the Completion3D dataset. The second row are the corrected pairs by minimizing the one-side CD loss.

the network training. Moreover, the scale-inconsistency issue also gives rise to unreliable evaluation results, as we expect the network to predict a complete point cloud of a consistent scale with the incomplete point cloud. Therefore, it is necessary to correct the scales of these pairs before using the dataset.

We leverage the one-side CD loss to identify and correct these pairs. For a consistent pair of the incomplete point cloud $C$ and complete point cloud $X$, the one-side CD loss should be very low:

$$\mathcal{L}_{\text{1-Side CD}}(C, X) = \frac{1}{|C|} \sum_{c \in C} \min_{x \in X} ||c - x||^2. \tag{8}$$

We find the correct scale of the incomplete point cloud by optimizing the following problem

$$\min_{\delta} \mathcal{L}_{\text{1-Side CD}}(\delta C, X). \tag{9}$$

This optimization problem is solved by using the python package scipy.optimize.fmin for every pair of point clouds in the Completion3D dataset. We consider the pairs with a scale factor $\delta$ greater than 1.05 or less than 0.95 as inconsistent pairs, and then correct its scale inconsistency by multiplying the scale factor $\delta$ to these incomplete point clouds. In the training set, we find $2.81\%$ pairs are inconsistent. The inconsistent pairs are also discovered in the validation set. We can not verify the test set because the ground truth complete point cloud is not released.

We did not use the online Completion3D benchmark server to evaluate our method and previous methods for the following reasons: 1) the website server was out of service, as it gave no feedback for any submissions at the time we conduct this work; 2) the ground truth complete point clouds in test set of the Completion3D dataset are not released, and hence we can not verify whether this inconsistency problem is also present in the test set. Therefore, we use the test set provided in the work (Wang et al., 2020), which contains 1200 pairs of incomplete-complete point clouds for testing. It contains the same set of objects as the test set of the original Completion3D dataset. After correcting inconsistent pairs in this test set, we evaluate our method and previous methods on this revised test set to achieve fair and reliable comparisons.

### B.5 COMPLETE EXPERIMENT RESULTS FOR MVP-40 DATASET

In Table 1 in the main text, we only present the completion result at the missing ratio of 50% and 12.5% for the MVP-40 dataset. We present the complete experiment result on MVP-40 dataset including result at the 25% missing ratio in Table 5 below.

Table 5: Complete Point cloud completion results on MVP-40 dataset. The missing ratio is at 50%, 25% and 12.5%, respectively. CD loss is multiplied by $10^4$. EMD loss is multiplied by $10^2$.

| Method | MVP40 (50% missing) | | | MVP40 (25% missing) | | | MVP40 (12.5% missing) | | |
|---|---|---|---|---|---|---|---|---|---|
| | CD | EMD | F1 | CD | EMD | F1 | CD | EMD | F1 |
| PCN (Yuan et al., 2018) | 39.67 | 6.37 | 0.581 | 34.40 | 6.21 | 0.606 | 32.56 | 6.18 | 0.619 |
| TopNet (Tchapmi et al., 2019) | 48.52 | 8.75 | 0.506 | 42.39 | 10.25 | 0.520 | 40.12 | 9.08 | 0.542 |
| FoldingNet (Yang et al., 2018) | 51.89 | 11.66 | 0.441 | 45.99 | 9.85 | 0.475 | 46.03 | 8.93 | 0.480 |
| MSN (Liu et al., 2020) | 34.33 | 9.70 | 0.646 | 23.14 | 6.59 | 0.712 | 20.20 | 4.54 | 0.728 |
| Cascade (Wang et al., 2020) | 34.16 | 15.40 | 0.635 | 29.13 | 8.16 | 0.647 | 26.73 | 5.71 | 0.657 |
| ECG (Pan, 2020) | 34.06 | 16.19 | 0.671 | 28.01 | 10.79 | 0.717 | 16.90 | 6.20 | 0.774 |
| GRNet (Xie et al., 2020) | 35.99 | 12.33 | 0.589 | 25.84 | 8.43 | 0.626 | 22.04 | 6.43 | 0.646 |
| PMPNet (Wen et al., 2021) | **25.41** | 29.92 | 0.721 | **15.73** | 16.08 | **0.815** | 13.00 | 8.92 | 0.815 |
| VRCNet (Pan et al., 2021) | 25.70 | 18.40 | 0.736 | 18.28 | 10.96 | 0.776 | 14.20 | 5.90 | 0.807 |
| PDR paradigm (Ours) | 27.20 | **2.68** | **0.739** | 16.54 | **1.68** | 0.800 | **12.70** | **1.39** | **0.827** |

## B.6 COMPLETE EXPERIMENT RESULTS FOR NETWORK ABLATION STUDY

Table 6: Comparison of coarse point clouds generated by conditional generation networks of different structures on MVP dataset at the resolution of $2048$ points. Experiments are conducted under two circumstances: with and without data augmentation. The networks without data augmentation are trained for 300 epochs, and networks with data augmentation are trained for 600 epochs. The data augmentation we use is specified in Table 4 for the MVP dataset. We report the networks' performance on both the training set and the test set. We can see the overfitting problem is largely mitigated by data augmentation.

| Model | Data Augmentation | CD | | EMD | | F1 | |
|---|---|---|---|---|---|---|---|
| | | Train | Test | Train | Test | Train | Test |
| Pointwise-net | ✗ | 6.96 | 11.99 | 0.88 | 1.63 | 0.328 | 0.265 |
| Concate $x^t$ & $c$ | ✗ | 9.96 | 10.79 | 1.57 | 1.54 | 0.397 | 0.382 |
| PointNet++ | ✗ | 6.44 | 9.39 | 0.84 | 1.38 | 0.397 | 0.355 |
| PA-Deconv | ✗ | 5.85 | 8.81 | 0.73 | 1.34 | 0.425 | 0.379 |
| PA-Deconv & Att. | ✗ | 5.56 | 8.71 | 0.70 | 1.29 | 0.443 | 0.389 |
| Pointwise-net | ✓ | 11.30 | 12.69 | 1.33 | 1.61 | 0.262 | 0.246 |
| Concate $x^t$ & $c$ | ✓ | 14.52 | 16.31 | 1.90 | 1.99 | 0.414 | 0.395 |
| PointNet++ | ✓ | 9.86 | 11.18 | 1.43 | 1.74 | 0.402 | 0.376 |
| PA-Deconv | ✓ | 9.22 | 10.78 | 1.24 | 1.50 | 0.407 | 0.381 |
| PA-Deconv & Att. | ✓ | 7.98 | 9.24 | 1.03 | 1.32 | 0.436 | 0.409 |

In Section 5.3 in the main text, we conduct ablation study of our proposed network architecture, and results are shown in Table 3. Note that the conditional generation networks in DDPM in Table 3 are trained without data augmentation. However, it is actually very important to train the conditional generation networks with data augmentation, because we need to prevent it from overfitting on the training set, so that they can generate coarse point clouds of consistent distribution on the training set and test set to train the refinement network. Therefore, we provide the training results with data augmentation in Table 6. The data augmentation is specified in Table 4 for MVP dataset.

Same as Table 3 in the main text, "PA-Deonv & Att." is our proposed complete network shown in Figure 2. "PA-Deonv" is our network without attention mechanism. "PointNet++" further removes the PA-Deconv module. "Concate $x^t$ & $c$" removes FT modules. It concatenates $c$ with $x^t$ as Zhou et al. (2021) do, and feed them to a single PointNet++ with attention mechanism and PA-Deconv. "Pointwise-net" only utilizes a global feature extracted from the incomplete point cloud. We can see that these proposed modules indeed improve the networks' performance. Our proposed networks achieve superior results both with and without data augmentation.

We also observe that networks generally achieve better performance on both the training set and test set without data augmentation, but they tend to overfit on the training set. This is undesirable because we need these conditional generation networks to generate training data for the refinement networks. It is very important for them to generate coarse point clouds that have consistent distributions on the training set and the test set. Indeed, we can see that the overfitting problem is largely mitigated in the presence of data augmentation.

### B.7 Complete Experiment Results For DDPM Acceleration

The complete experiment results of the DDPM acceleration is shown in Table 7. We can see that the quality of coarse point clouds generated by the accelerated DDPMs has dropped considerably. However, with the help of the Refinement Network, the performance drop of the final refined point clouds is slight. This demonstrates the strong refinement capability of our proposed network architecture shown in Figure 2 in the main text.

Table 7: Refine coarse point clouds generated by the accelerated DDPMs on the MVP dataset at the resolution of $2048$ points. We can see performance drop is slight for the refined point clouds. We also report the average generation time of a single point cloud evaluated on one NVIDIA GEFORCE RTX 2080 Ti GPU for DDPM of different reverse steps.

| Number of Reverse Steps | Average Generation Time | CD | | EMD | | F1 | |
|---|---|---|---|---|---|---|---|
| | | Coarse | Refined | Coarse | Refined | Coarse | Refined |
| 1000 (Original) | 16.86 s | 10.69 | 5.66 | 1.46 | 1.37 | 0.400 | 0.499 |
| 50 | 0.78 s | 13.19 | 5.68 | 1.65 | 1.47 | 0.341 | 0.493 |
| 20 | 0.32 s | 18.12 | 5.78 | 1.99 | 1.56 | 0.255 | 0.474 |

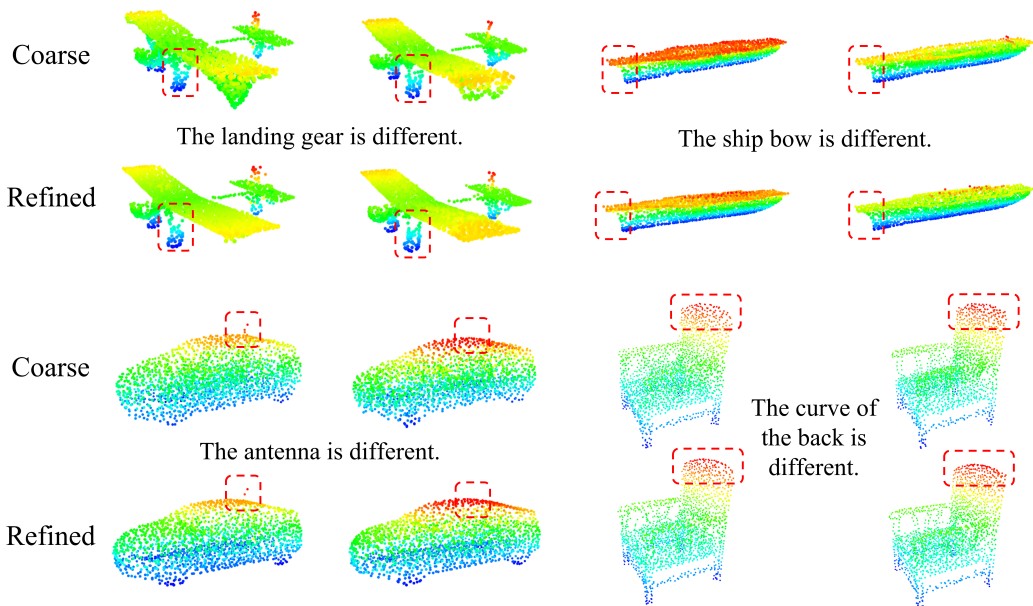

Figure 8: Our PDR paradigm demonstrates diversity in the completion results. For each object, the two images in the first row are coarse completion results from a trained DDPM generated in two trials for the same incomplete point cloud. The two images in the second row are refined results for the two coarse point clouds, respectively. We can see that some diversity is preserved after the refinement.

## B.8 GENERATION DIVERSITY OF THE PDR PARADIGM

In this section, we discuss whether the PDR Paradigm can generate diverse completion results for the same incomplete point cloud. Although there is no stochasticity in the refinement network, we find our PDR paradigm still demonstrates some kind of diversity in the completion results.

We know that DDPM itself is a probabilistic model and can generate diverse completion results. The refinement network receives a coarse completion from the DDPM and then refines it according to the condition point cloud, *i.e.*, the incomplete point cloud. The final refined result surely depends on the condition point cloud, but also depends on the coarse point cloud received from the DDPM. The refinement network can only refine the coarse point cloud in a small scale, because we multiply the output of the refinement network by a small constant $\gamma = 0.001$ as described in Section 3.3 in the main text. Therefore, the overall sketch of the coarse point cloud will be preserved after the refinement. This explains why the PDR paradigm still bears low EMD loss as the DDPM, even though we use CD loss to train the refinement network, because the refinement network does not change the overall distribution of the coarse shape generated by DDPM.

Back to the diversity issue, if the coarse completion results from DDPM demonstrate diversity for the same incomplete point cloud, the refined point clouds will also demonstrate some diversity because the inputs to the refinement network are different. Figure 8 shows some examples where the PDR paradigm demonstrate diversity in the completion results.

### B.9 MORE VISUALIZATION RESULTS

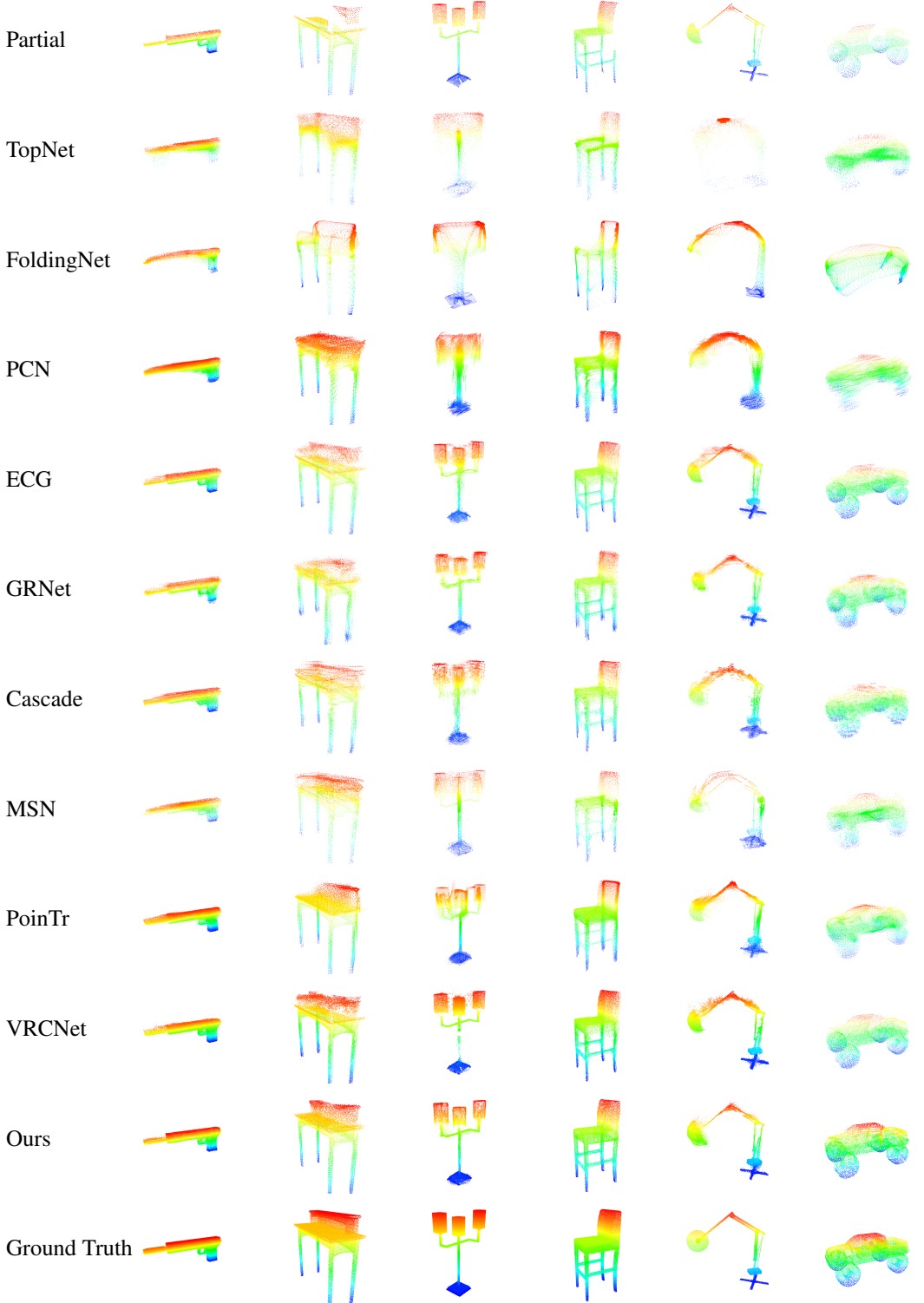

Figure 9: Visual comparison of our method and other baselines. Samples are from the MVP dataset at the resolution of 16384 points. We can see that point clouds generated by our method generally have better visual quality.

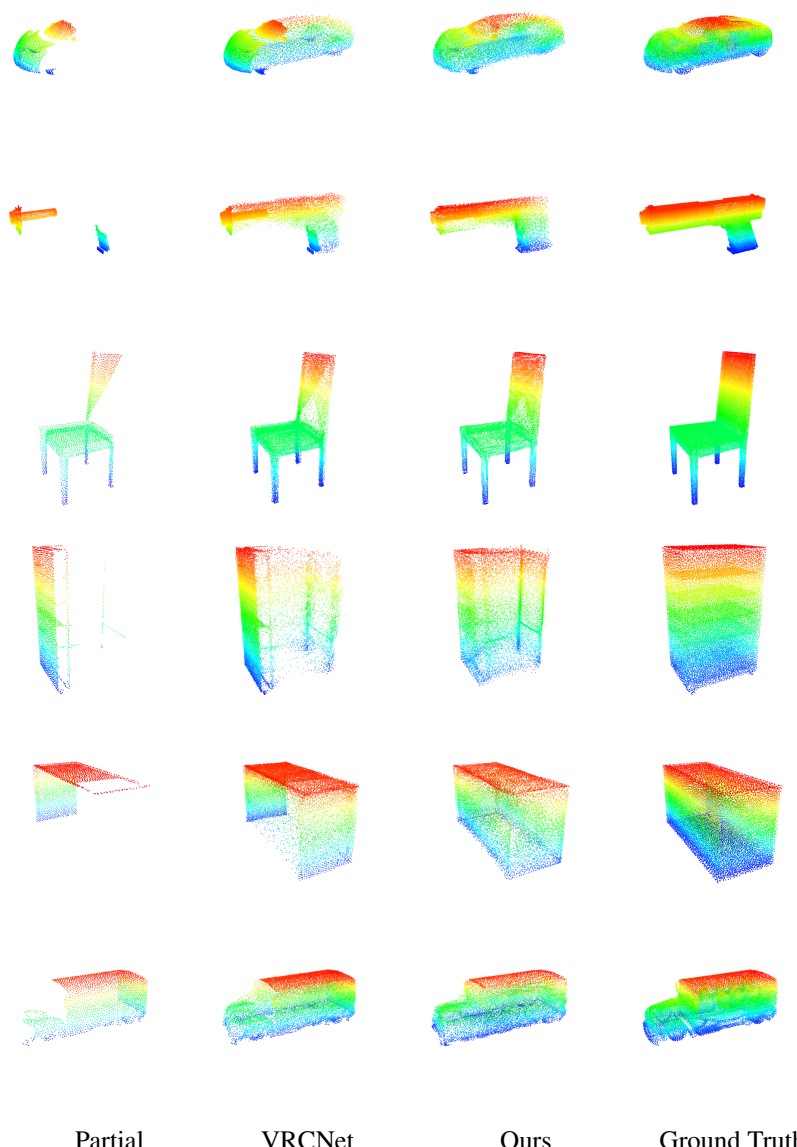

Partial        VRCNet        Ours        Ground Truth

Figure 10: Visual comparison of our method and VRCNet. Samples are from the MVP dataset at the resolution of 16384 points. We can see that VRCNet sometimes tend to predict more points to the parts that are known in the incompelte point cloud, while put less points at the missing part. This could effectively reduce CD loss, but leads to large EMD loss. Compared with VRCNet, our method generally generates more uniform point clouds.

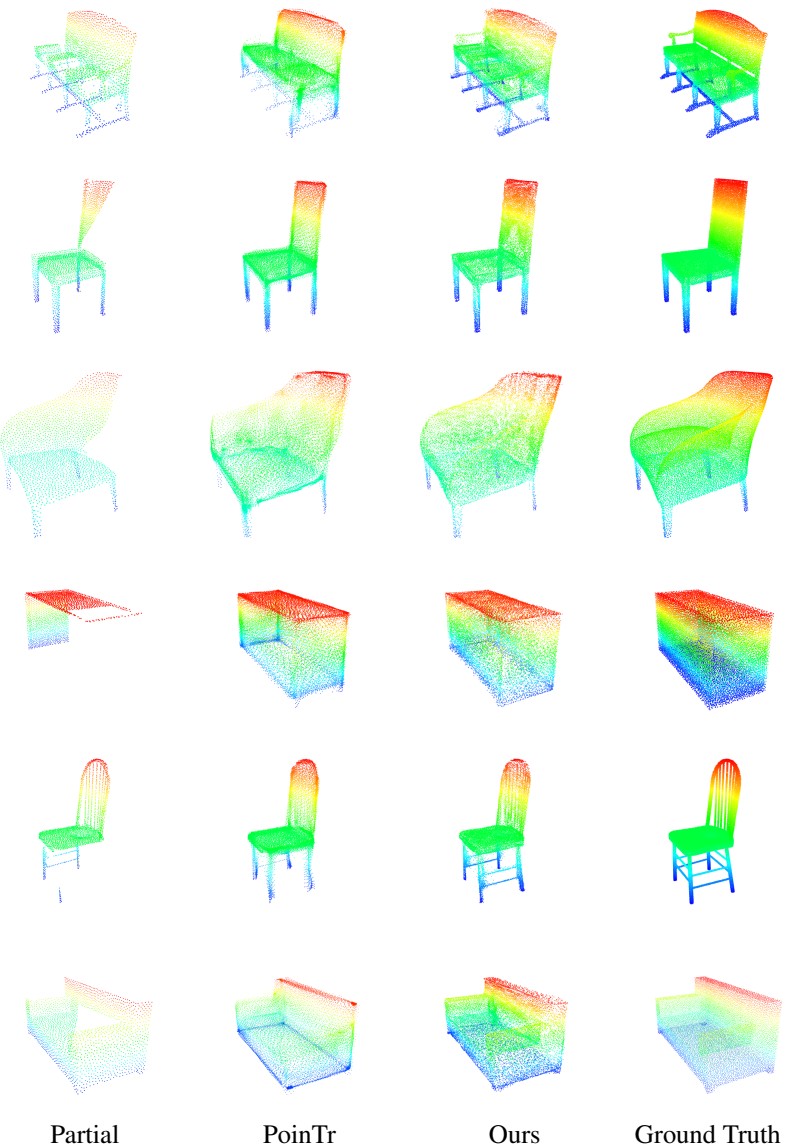

Partial  PoinTr  Ours  Ground Truth

Figure 11: Visual comparison of our method and PoinTr. Samples are from the MVP dataset at the resolution of 16384 points. We can see that PoinTr sometimes tend to predict more points at the skeleton of objects, while points on surfaces seem sparse. Compared with PoinTr, our method generally generates more uniform point clouds.

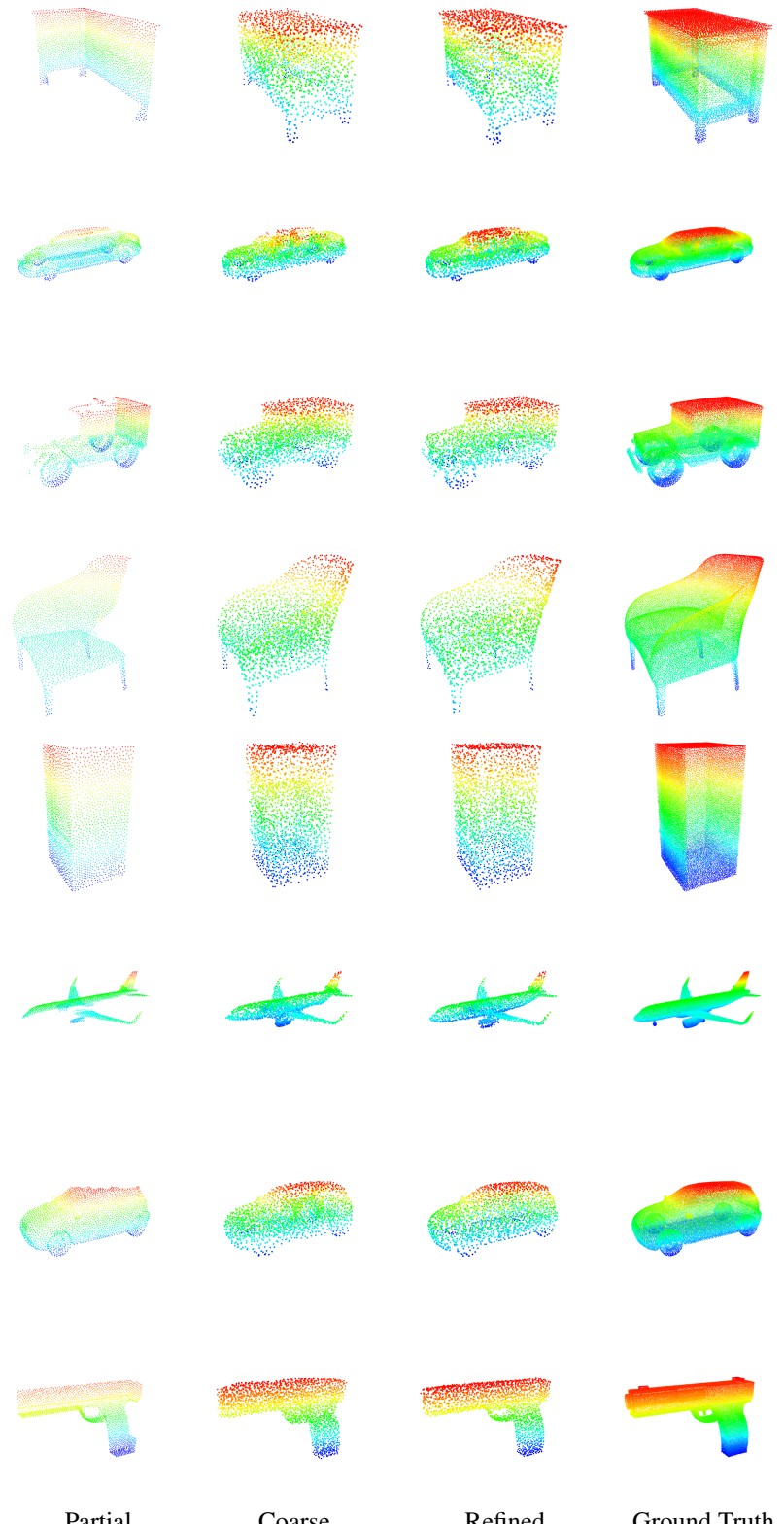

Partial          Coarse          Refined          Ground Truth

Figure 12: Visual comparison of coarse point clouds generated by the Conditional Generation Network in DDPM and point clouds after refinement. Samples are from the MVP dataset at the resolution of 2048 points. We can see that coarse point clouds generated by the Conditional Generation Network basically uniformly cover the overall shape of objects, but tend to be noisy. After refinement, point clouds demonstrate both good overall density distribution and sharp local details.

