# OpenReview forum: "A Conditional Point Diffusion-Refinement Paradigm for 3D Point Cloud Completion"
_ICLR.cc/2022/Conference — ICLR 2022 Poster_

### Official Review · Reviewer_bEQt · 2021-10-30

**Correctness:** 3
**Technical Novelty And Significance:** 3
**Empirical Novelty And Significance:** 3
**Recommendation:** 6
**Confidence:** 5

**Main Review:**

**Strengths**

1. The manuscript is overall clearly written.
2. The authors propose a novel pipeline for point cloud completion.

**Weaknesses**

1. The figures in the manuscript are not clear enough. For example, the relationship between CGNet and DDPM is not clearly represented in Figure 1. The relationship of CGNet, RFNet, CFENet, and Denoise Net are not well represented. This unclarity makes me wonder that both CGNet and RFNet have CFENet and Denoise Net, but why do they have different names?
2. The authors claim that the generated results are more uniform than other methods. Please provide evidence.
3. It seems that the results of the Completion3D benchmark in Table 1 are not consistent with the online results [here](https://completion3d.stanford.edu/results). Could you explain the reason?
4. There are lots of methods in Tables 1-3. However, there are only a few methods in Figure 4. Please provide more qualitative results in Figures 4.
5. Please provide the running time comparison with other methods because you incoporate lots of time-comsuming operations.

**Summary Of The Paper:**

In this paper, the authors propose a novel Point Diffusion-Refinement (PDR) paradigm for point cloud completion. The proposed method effectively and efficiently extracts multi-level features from partially observed point clouds to guide completion. Moreover, it accurately manipulates spatial locations of 3D points to obtain smooth surfaces and sharp details. Experimental results demonstrate its state-of-the-art performance on the mainstream datasets and benchmarks.

**Summary Of The Review:**

I'm leaning to reject this paper (Borderline Reject) *at this time* due to the following reasons:

1. Some claims in the manuscript lacks references.
2. The results of the Completion 3D benchmark are not consistent with the online results.

**Update (11/26/2021)**

The authors address all of my concerns in the rebuttal. Therefore, I agree to accept the paper.

---

> ### Author Response · Authors · 2021-11-19
> **Rebuttal to Reviewer bEQt**
>
> We thank the reviewer for the time to review our paper and valuable feedback. We have updated the paper. Please re-download the PDF file for reference.
>
> 1.Explain CGNet, RFNet, Condition Feature Extraction subnet, Denoise subnet.
>
> Both the Conditional Generation network (CGNet) and the Refinement network (RFNet) have the same architecture: A dual-path architecture shown in Figure 2 (see caption of Figure 2 and the beginning of Section 3.3). The dual-path architecture contains two subnets: the Condition Feature Extraction subnet and the Denoise subnet. The Condition Feature Extraction subnet extracts multi-level features from the condition (incomplete) point cloud. The Denoise subnet utilizes these features to move the input noisy point cloud towards a clean and complete point cloud.
>
> The CGNet and the RFNet have different names because they are designed for different goals with different losses. The CGNet iteratively moves a Gaussion noise towards a complete but coarse point cloud according to the DDPM framework. It is a parameterization of the function \epsilon_{\theta} described in the paragraph before Equation 4 and the beginning of Section 3.2. It is trained to minimize DDPM loss in Equation 4. The RFNet refines the coarse point cloud generated by the CGNet. The RFNet is trained to minimize CD loss in Equation 5.
>
> 2.Uniformness.
>
> Quantitatively, the uniformness of our PDR paradigm is reflected by its low EMD loss compared with other methods. EMD is highly indicative of uniformness as it conducts bijective matching of points between the generated point cloud and the ground truth point cloud. Since the ground truth point cloud is uniform, a model has to generate the same uniform point cloud to achieve low EMD loss. This point of view is also mentioned in the paper "CVPR 2021 Unsupervised 3D Shape Completion through GAN Inversion".
>
> Qualitatively, we provide some visual results in Figure 10 and 11 in the appendix. We can see that our PDR paradigm can generate more uniform point clouds than VRCNet and PoinTr, which are two strongest baselines.
>
> 3.Different from Online Completion3D benchmark.
>
> We do not evaluate our PDR paradigm or directly cite the results of other baselines on the Completion3D benchmark website. Instead, we use the revised test set provided in the work "CVPR 2020 Cascaded refinement network for point cloud completion" for fair and reliable comparison. There are three reasons for this.
>
>     1) We retrain all baselines with our proposed data augmentation described in Appendix Section B.3 for fair comparison. We observe that data augmentation can improve their performance.
>
>     2) We are unable to evaluate our PDR paradigm on the Completion3D benchmark website because the website server was out of service at the time of this work. We were also unable to contact the authors.
>
>     3) Evaluation results from the Completion3D benchmark website may not be accurate. We find many pairs of incomplete-complete point clouds have inconsistent scales in both the training set and validation set. See Figure 7 in the appendix for a few examples. We correct these inconsistent pairs in the training set before training our PDR paradigm and other baselines. We cannot verify if this problem exists in the test set because the ground truth complete point clouds for the test set are not released.
>
> Therefore, we use the test set provided in the work "CVPR 2020 Cascaded refinement network for point cloud completion", which contains 1200 pairs of incomplete-complete point clouds for testing. It contains the same set of objects as the test set of the original Completion3D dataset, and the ground truth complete point clouds are released. After correcting inconsistent pairs in this test set, we evaluate our method and previous methods on this revised test set to achieve fair and reliable comparisons. We have added a section to discuss this issue in Appendix B.4.
>
> 4.Figure 4.
>
> We only provide the visual results for a few strongest baselines in Figure 4 because of page limits. More visual results for all baselines are demonstrated in Figure 9 in the appendix.
>
> 5.Running time of DDPM.
>
> The average generation time of a single point cloud of our trained 1000-step, 50-step, 20-step DDPMs are 16.86 s, 0.78 s, 0.32 s evaluated on a single NVIDIA GEFORCE RTX 2080 Ti GPU. If we replace our dual path network with the Pointwise net, Vanilla Pointnet ++, or Point Voxel CNN, the average generation time of a 1000-step DDPM becomes 1.28 s, 5.27s, and 12.77 s, respectively. Although our dual-path network is slower than them, we achieve better performance than them as shown in Table 3. It is a trade-off between computational cost and generation quality.
> See Point 3 of our response to Reviewer qA73 for more discussions on accelerating the iterative generation process of DDPM.

---

### Official Review · Reviewer_qA73 · 2021-10-31

**Correctness:** 4
**Technical Novelty And Significance:** 4
**Empirical Novelty And Significance:** 3
**Recommendation:** 8
**Confidence:** 4

**Main Review:**

Pros:
State-of-the-art performer for point cloud completion on various datasets.
I am impressed by the comprehensiveness of the paper. All the key details and experiments are included and well-presented.
Furthermore, since the specific problem of point cloud completion using a DDPM is a relatively under-studied problem, there are many design decisions that need to be made and justified. The paper is very good in this respect.
Various components in the original PointNet++, such as up- and down-sampling modules, are carefully analyzed of their inner workings, taylored to the proposed setting, and evaluated with ablation studies.

Questions:
1. In some sense, the refinement network can be considered as one extra step to the DDPM generator. I wonder if you have any insight on how does the proposed two-stage generation + refinement paradigm compare with using a single stage DDPM, but with a larger model?

2. in section B.1, the author mentiond that the reported results are evaluated on a subset due to the slow evaluation speed. Since the confidence intervals of the evaluation metrics w.r.t. subsampling are not reported, I wonder if subsampling the test set can potentially lead to unreliable and unfair comparison with previous methods?

3. Regarding evaluation speed, I wonder how long does it take to evaluate one point cloud?

4. Regarding Table 5, is the refinement network retrained for experiments with different DDPM steps?

**Update:**
Thank the author for providing additional insights on how the 2-staged model can improve the result -- indeed, minimizing ELBO does not imply minimizing Chamfer distance. I am also glad to learn that the comparisons with baselines are fair. I thus retain my rating of accept.


**Summary Of The Paper:**

The paper tackles the problem of shape-level point cloud completion in a fully supervised setting. The big picture of the proposed method is to use a conditional DDPM to generate a noisy but complete point cloud given incomplete input, and then use another refinement network, also conditional on the incomplete input, to further refine the noisy point cloud. Both networks adopt a novel, dual-path network architecture, based on PointNet++, to allow localized guidance from the incomplete point cloud. Thanks to the 2-staged approach, up to 50x acceleration of evaluation speed is achievable using step jumping, with moderate performance drop. The model is evaluated on MVP, MVP-40 and Completion3D datasets.

**Summary Of The Review:**

Good performance. Novel approach. Comprehensive paper. Contributions are well justified. some concern on the reliability of the reported numbers due to subsampling the test set.

---

> ### Author Response · Authors · 2021-11-19
> **Rebuttal to Reviewer qA73**
>
> We thank the reviewer for the time to review our paper and valuable feedback. We have updated the paper. Please re-download the PDF file for reference.
>
> 1.Refinement network can be considered as one extra step to the DDPM.
>
> That's a very great point and exactly what we want to explore in future work. Indeed, we can consider the refinement step as an extra step (the 0-th step) in the reverse process of the DDPM.
>
> As shown in Figure 1 and Figure 12, point clouds generated by DDPM are often noisy, lack smooth surfaces and sharp details. We think there are two possible reasons: 1) The probabilistic nature of the generation process of DDPM. 2) The training loss of DDPM is to maximize the evidence-lower bound (ELBO) of the log-likelihood of training data. Although ELBO works well for 2D images, its inductive bias is unclear in the 3D domain.
>
> Therefore, we think it may not be helpful to simply increase the size of the conditional generation network in the DDPM. Instead, we need another network to directly minimize the discrepancy between DDPM outputs and ground truth point clouds.
>
> On the other hand, we do think it is a great idea to combine the refinement network and conditional generation network as a whole. That is, we can train the two networks jointly in an end-to-end fashion, instead of 2 separate stages. We leave this for future work.
>
> 2.Evaluation method.
>
> The comparison with previous methods is reliable and fair. Subsampling is used only when choosing the best checkpoint for the conditional generation network in DDPM. We then use it to generate training data for the refinement network. The whole PDR paradigm (composed of the conditional generation network and refinement network) and previous methods are evaluated on the complete test set. Therefore, the comparison result in Table 1 and Table 2 is reliable and fair. We have added a clarification in Appendix B.1.
>
> 3.Generation time of DDPM.
>
> The average generation time of a single point cloud of our trained 1000-step, 50-step, 20-step DDPMs are 16.86 s, 0.78 s, 0.32 s evaluated on a single NVIDIA GEFORCE RTX 2080 Ti GPU. Most of the computation is in the iterative generation process. It is still an active research field to accelerate the generation process of DDPM. We notice that a recent paper (https://openreview.net/forum?id=TIdIXIpzhoI) has reduced the number of generation steps to as few as 4 in the image domain. We think our DDPM for 3D point cloud generation can be further accelerated using techniques proposed by new research on DDPM acceleration.
>
> 4.Is the refinement network retrained for experiments with different DDPM steps?
>
> Yes. We retrain new refinement networks for DDPMs with different generation steps.

---

### Official Review · Reviewer_prns · 2021-11-04

**Correctness:** 4
**Technical Novelty And Significance:** 4
**Empirical Novelty And Significance:** 3
**Recommendation:** 8
**Confidence:** 5

**Main Review:**

I thoroughly enjoyed this paper. Though the paper casts itself as a "point cloud completion" paper, I find that the architecture presented is more general and more convincingly powerful than previous DDPM models proposed for general 3D point cloud generation (Luo et al, CVPR'21 and Zhou et al, ICCV'21).

Luo's approach, which was a best paper finalist for CVPR, is comparatively much weaker in its conditionalization strategy, whereas this proposed work utilizes a multi-scale dual U-Net type approach which allows multi-scale conditioning information to propagate through the network, which seems to add to additional fidelity. Zhou's approach uses point-voxel CNN's, making the (mostly empirical) argument that some kind of point ordering (i.e. voxelization) is necessary for diffusion and so something like PointNet++ cannot be used. This argument never sat well with me, so I was happy to see the authors in this paper also rebut Zhou's reasoning, and additionally present an easy and elegant fix-- to add the absolute coordinates to the PointNet feature (similar in spirit to the CoordConv approach of Liu et al, NeurIPS'18).

The paper is well-presented and easy-to-read, with a robust supplementary. All design choices were reasonably discussed, and I was happy to see that the authors stressed how Chamfer distance losses could be avoided using the DDPM design, a point which isn't always well-explained in the point cloud diffusion literature (even though the authors end up using CD-based loss anyway in the refinement module). I like and agree with the proposed minor modifications to the PointNet++ modules, as well as the use of self-attention instead of pooling. I'm not 100% sold on the refinement network-- I think the training procedure of generating 10 coarse models seems a bit ad hoc-- but I do think it represents an interesting trade-off, and I'm glad the authors included it.

Here's my wishlist for this paper:
1. Direct comparisons against Point-Voxel diffusion completion results. This is the most apples-to-apples comparison you could make.
2. Multimodal completions. My guess is that the refinementNet probably hurts your ability to get diverse completion results? Perhaps with some stochasticity in the feature transfer module could be added to produce multimodal completions? If your method is unable to provide multimodal completions, you should discuss this limitation.
3. Representation learning results. Recently it was shown that point cloud completion can be a powerful pretext task for representation learning on point clouds (see Wang et al, "Unsupervised Point Cloud Pre-training via Occlusion Completion" ICCV'21). I would be very curious what would happen if you attempted to extract a representation from your trained feature transfer modules and use it for a different downstream supervised task. My guess is that it could very well outperform previous art.
4. "In the wild" completion results: for example, training on shapenet/modelnet cars and then conditioning from bounding box crops of partially occluded car point clouds in KITTI
5. Point cloud generation with depth map conditioning modality (like as done in Point-Voxel diffusion)

I caught a few typos. Please comb over the paper one more time. For example, from the first couple pages I see: "for many downstreaming applications" and "it can iteratively moves a set of Gaussian noise" and "we propose the feature transform (FT) module to directly transmits encoded point features"

**Summary Of The Paper:**

This paper proposed a novel way to utilize DDPM's for point cloud completion. They use a conditional DDPM to generate a plausible "complete" point cloud from pure Gaussian noise, conditional on a partial subset of the point cloud points. Their main contribution is is in the design of the conditional feature extraction and denoising subnet, which essentially form dual path connected U-Net type structures with internal modules based on an improved PointNet++ design. Additionally, the authors propose a refinement network to mitigate the computational burden of needing so many diffusion steps, which seems to offer an acceptable performance tradeoff for increases computational efficiency. The method convincingly demonstrates superior performance over its competitors, with crisp details preserved on the datasets tested.

**Summary Of The Review:**

Overall, I really like this paper. I would like to see the authors address the items in my wishlist (specifically iterms #1-3), but the paper stands pretty well on its own.

I also think that diffusion models and 3D point clouds is a nice and timely area of research, and this paper will likely be a nice reference point for others trying to combine DDPM's with 3D point cloud applications.

---

> ### Author Response · Authors · 2021-11-19
> **Rebuttal to Reviewer prns**
>
> We thank the reviewer for the time to review our paper and valuable feedback. We have updated the paper. Please re-download the PDF file for reference.
>
> 1.Direct comparisons against Point-Voxel diffusion completion results.
>
> We think this is a great suggestion. The code for the Point-Voxel diffusion model was not released until recently. We included additional experiments to compare with their method on the MVP dataset at the resolution of 2048 points. Their method concatenates the condition point cloud c with the noisy point cloud x^t, and feed them to a single point-voxel CNN.
>
> We train the point-voxel CNN for 300 epochs. Its CD loss = 12.97E-4, EMD loss = 1.82E-2, and F1 score = 0.331. If we replace the point-voxel CNN with our improved PointNet++, the CD loss = 10.78E-4, EMD loss = 1.54E-2, and F1 score = 0.382 (see Table 3). This means our improved PointNet++ itself is better than the point-voxel CNN in terms of generating complete point clouds.
>
> In addition, if we further use the dual-path conditional generation network in Figure 2 and include the refinement network, our complete PDR paradigm achieves even better result than the Point-Voxel diffusion model: Our PDR paradigm has CD loss = 5.66E-4, EMD loss = 1.37E-2, and F1 score = 0.499 (see Table 1).
>
> 2.Ability to get diverse completion results.
>
> We think this is an important aspect to discuss. It is true that there is no stochasticity in the refinement network, but we find our PDR paradigm still demonstrates diversity in the completed point clouds. This is because DDPM is a probabilistic (and therefore stochastic) model and can generate diverse coarse point clouds. The refinement network receives a coarse completion from the DDPM and then refines it according to the conditional (incomplete) point cloud. Because the refinement network can only slightly modify the coarse point cloud (since there is a small constant factor gamma=0.001 as described in Section 3.3), the overall sketch of the coarse point cloud is preserved after refinement. We added the discussion in Appendix B.8, which also includes several examples of the diversity in PDR paradigm's completion results.
>
> We also think it is a great suggestion to explicitly add some stochasticity to the refinement network, so that the PDR paradigm can have more diversity in generation. This can hopefully be achieved by using some techniques in variational auto-encoders, and we leave it for future work.
>
> 3~5. Additional experiments.
>
> We think these are indeed great suggestions. Due to the time limit, we are unable to conduct experiments on this part. We will try to have some preliminary experiments by camera-ready if the paper is accepted, and leave systematic study to future work.
>
> 6.Typos.
>
> Thanks for your kind reminder. We have fixed them accordingly.

---

### Decision · Program_Chairs · 2022-01-20

**Decision:**

Accept (Poster)

**Comment:**

This paper presents an approach based on conditional denoising diffusion models for point cloud completion. The reviewers have recognized the significance of contributions, the clarity of presentation, and the comprehensivity of experiments. I am happy to recommend this paper for presentation at ICLR.